# How Robust is Unsupervised Representation Learning to Distribution Shift?

**Yuge Shi**[*]
Department of Engineering Science
University of Oxford

**Imant Daunhawer & Julia E. Vogt**
Department of Computer Science
ETH Zurich

**Philip H.S. Torr**
Department of Engineering Science
University of Oxford

**Amartya Sanyal**
Department of Computer Science & ETH AI Center
ETH Zurich

## ABSTRACT

The robustness of machine learning algorithms to distributions shift is primarily discussed in the context of supervised learning (SL). As such, there is a lack of insight on the robustness of the representations learned from unsupervised methods, such as self-supervised learning (SSL) and auto-encoder based algorithms (AE), to distribution shift. We posit that the *input-driven* objectives of unsupervised algorithms lead to representations that are more robust to distribution shift than the *target-driven* objective of SL. We verify this by extensively evaluating the performance of SSL and AE on both synthetic and realistic distribution shift datasets. Following observations that the linear layer used for classification itself can be susceptible to spurious correlations, we evaluate the representations using a linear head trained on a small amount of out-of-distribution (OOD) data, to isolate the robustness of the learned representations from that of the linear head. We also develop "controllable" versions of existing realistic domain generalisation datasets with adjustable degrees of distribution shifts. This allows us to study the robustness of different learning algorithms under versatile yet realistic distribution shift conditions. Our experiments show that representations learned from unsupervised learning algorithms generalise better than SL under a wide variety of extreme as well as realistic distribution shifts.

## 1 INTRODUCTION

Machine Learning (ML) algorithms are classically designed under the statistical assumption that the training and test data are drawn from the same distribution. However, this assumption does not hold in most cases of real world deployment of ML systems. For example, medical researchers might obtain their training data from hospitals in Europe, but deploy their trained models in Asia; the changes in conditions such as imaging equipment and demography result in a shift in the data distribution between train and test set (Dockès et al., 2021; Glocker et al., 2019; Henrich et al., 2010). To perform well on such tasks requires the models to generalise to unseen distributions — an important property that is not evaluated on standard machine learning datasets like ImageNet, where the train and test set are sampled i.i.d. from the same distribution.

With increasing attention on this issue, researchers have been probing the generalisation performance of ML models by creating datasets that feature distribution shift tasks (Koh et al., 2021; Gulrajani and Lopez-Paz, 2020; Shah et al., 2020) and proposing algorithms that aim to improve generalisation performance under distribution shift (Ganin et al., 2016; Arjovsky et al., 2019; Sun and Saenko, 2016; Sagawa et al., 2020; Shi et al., 2022). In this work, we identify three specific problems with current approaches in distribution shift problems, in computer vision, and develop a suite of experiments to address them.

---

[*]Corresponding author, `yshi@robots.ox.ac.uk`

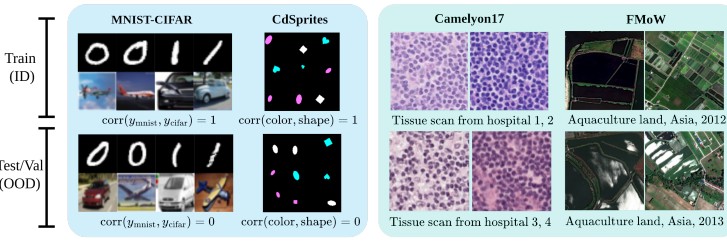

Figure 1: **Synthetic vs. realistic distribution shift:** The distribution shift in synthetic datasets (*left*, MNIST-CIFAR and CdSprites) are usually extreme and controllable (adjusted via changing the correlation); for realistic datasets (*right*, WILDS-Camelyon17 and FMoW) distribution shift can be subtle, hard to identify and impossible to control.

## 1.1 EXISTING PROBLEMS AND CONTRIBUTIONS

**Problem 1: The outdated focus on supervised regime for distribution shift**    In ML research, distribution shift has been studied in various contexts under different terminologies such as simplicity bias (Shah et al., 2020), dataset bias (Torralba and Efros, 2011), shortcut learning (Geirhos et al., 2020), and domain adaptation and generalisation (Koh et al., 2021; Gulrajani and Lopez-Paz, 2020). Most of these work are carried out under the scope of supervised learning (SL), including various works that either investigate spurious correlations (Shah et al., 2020; Hermann and Lampinen, 2020; Kalimeris et al., 2019) or those that propose specialised methods to improve generalisation and/or avoid shortcut solutions (Arjovsky et al., 2019; Ganin et al., 2016; Sagawa et al., 2020; Teney et al., 2022). However, recent research (Shah et al., 2020; Geirhos et al., 2020) highlighted the extreme vulnerability of SL methods to spurious correlations: they are susceptible to learning only features that are irrelevant to the true labelling functions yet highly predictive of the labels. This behaviour is not surprising given SL's ***target-driven*** objective: when presented with two features that are equally predictive of the target label, SL models have no incentive to learn both as learning only one of them suffices to predict the target label. This leads to poor generalisation when the learned feature is missing in the OOD test set.

On the other hand, in recent times, research in computer vision has seen a surge of unsupervised representation learning algorithms. These include self-supervised learning (SSL) algorithms (e.g., Chen et al. (2020a); Grill et al. (2020); Chen and He (2021)), which learn representations by enforcing invariance between the representations of two distinctly augmented views of the same image, and auto-encoder based algorithms (AE) (Rumelhart et al., 1985; Kingma and Welling, 2014; Higgins et al., 2017; Burda et al., 2016), which learn representations by reconstructing the input image. The immense popularity of these methods are mostly owed to their impressive performance on balanced in-distribution (ID) test datasets — how they perform on distribution shift tasks remains largely unknown. However, in distribution shift tasks, it is particularly meaningful to study unsupervised algorithms. This is because, in comparison to SL, their learning objectives are more ***input-driven*** i.e. they are incentivised to learn representations that most accurately represent the input data (Chen et al., 2020a; Alemi et al., 2017). When presented with two features equally predictive of the labels, unsupervised learning algorithms encourage the model to go beyond learning what's enough to predict the label, and instead focus on maximising the mutual information between the learned representations and the input. We hypothesise that this property of unsupervised representation learning algorithms helps them avoid the exploitation of spurious correlations, and thus fare better under distribution shift, compared to SL.

***Contribution****: Systematically evaluate SSL and AE on distribution shift tasks.*  We evaluate and compare the generalisation performance of unsupervised representation learning algorithms, including SSL and AE, with standard supervised learning. See section 2 for more details on our experiments.

**Problem 2: Disconnect between synthetic and realistic datasets**    Broadly speaking, there exists two types of datasets for studying distribution shift: synthetic datasets where the shift between train/test distribution is explicit and controlled (e.g. MNIST-CIFAR (Shah et al., 2020), CdSprites (Shi et al., 2022)) and realistic datasets featuring implicit distribution shift in the real world (e.g. WILDS (Koh et al., 2021)). We provide visual examples in fig. 1.

Synthetic datasets allow for explicit control of the distribution shift and are, thus, an effective diagnostic tool for generalisation performance. However, the simplistic nature of these datasets poses

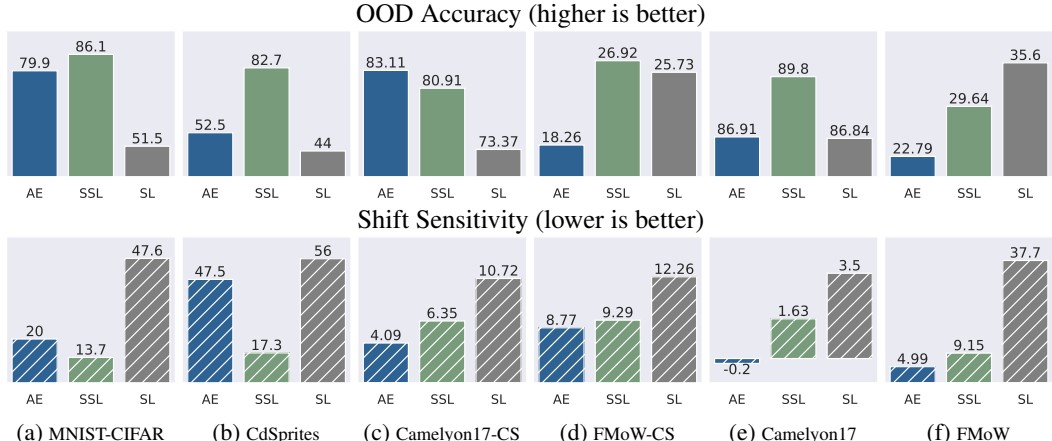

Figure 2: Performance of auto-encoder (AE), self-supervised learning (SSL), supervised learning (SL) models. **Top row**: the OOD test set accuracy (%) using linear heads trained on OOD data; **Bottom row**: Shift Sensitivity (see section 2 for definition), measures models' sensitivity to distribution shift. Note here $r_{\text{id}} = 1$ for Camelyon17-CS, FMoW-CS, and CdSprites (see sections 3.1.2 and 3.2.2).

concerns about the generality of the findings drawn from these experiments; a model's robustness to spurious correlation on certain toy datasets is not very useful if it fails when tested on similar real-world problems. On the other hand, realistic datasets often feature distribution shifts that are subtle and hard to define (see fig. 1, right). As a result, generalisation performances of different algorithms tend to fluctuate across datasets (Koh et al., 2021; Gulrajani and Lopez-Paz, 2020) with the cause of said fluctuations remaining unknown.

***Contribution***: *Controllable but realistic distribution shift tasks.* In addition to evaluating models on both synthetic and realistic datasets, we subsample realistic domain generalisation datasets in WILDS to artificially inject explicit spurious correlations between domains and labels. This allows us to directly control the level of shift in these realistic distribution shift datasets. We refer to them as *Controllable-Shift (CS) datasets.* [1]

**Problem 3: The linear classifier head strongly biases the evaluation protocol**   The most popular evaluation protocol for representation learning algorithms (both supervised and unsupervised) is *linear probing*. This involves freezing the weights of the representation learning model and training a linear classifier head on top of that to predict the labels. For distribution shift problems, this linear classifier is typically trained on the same training set as the representation learning backbone. Kang et al. (2020); Menon et al. (2020) observed the interesting phenomenon that this final layer linear classifier can be extremely susceptible to spurious correlations, causing poor OOD performance. Under simple, synthetic set up, they showed that SL models' performance on OOD test set can be dramatically improved by simply retraining the linear classifier on data where the spurious correlation is absent. This indicates that the linear classifier can be a strong source of model bias in distribution shift tasks, and to disentangle the linear classifier bias from the generalisation performance of the learned representations, it is advisable to re-train the linear head on OOD data during evaluation. Note that although retraining linear classifier head is already standard practice in transfer learning, its application is necessary as the pre-training task and target task are typically different; on the other hand, retraining linear head is neither necessary nor standard practice in distribution shift problems, despite the recognition of linear head bias in recent work (Kang et al., 2020; Menon et al., 2020).

***Contribution***: *OOD linear head.* When reporting OOD accuracy, we use a linear head trained on small amount of left-out OOD data as opposed to ID data, as is standard practice. This allows us to isolate the bias of the linear head from the generalisability of learned representations. We also quantify the linear head bias to highlight the importance of this treatment. With these results, we wish to establish OOD linear head evaluation as a standard protocol for evaluating robustness of representation learning algorithms to distribution shift.

---

[1]While datasets like ImageNet-9 and SVSF also provides distribution shift controls for real images, our sub-sampling of realistic domain generalisation datasets allows us to gain such control *cheaply* on a *large* set of datasets, and in addition provide further analysis on models' performance on *existing benchmarks* (WILDS).

In summary, we develop a suite of experiments and datasets to evaluate the performance of various representation learning paradigms under distribution shift. Figure 2 provides a summary of our results, comparing a range of methods from the following classes of algorithms: (i) SSL, (ii) AE, and (iii) SL. Note that though the intuition that unsupervised objectives should be better at distribution shift tasks is well-established in theory (Chen et al., 2020a; Alemi et al., 2017), state-of-the-art methods are predominantly developed under SL. To the best of our knowledge, we are the first to systematically evaluate and compare unsupervised representation learning methods to SL under distribution shift. The models are evaluated on both synthetic and realistic distribution shift datasets. Further, the models are also evaluated on CS datasets that contains controllable, explicit spurious correlations in realistic datasets. The main takeaways from this paper are:

- **SSL and AE are more robust than SL to extreme distribution shift:** Figures 2a to 2d shows results on distribution shift scenarios where the training set encodes extreme spurious correlations. In this setting, for both synthetic (figs. 2a and 2b) and real world (figs. 2c and 2d) datasets, SSL and AE consistently outperforms SL in terms of OOD accuracy (top row);
- **Compared to SL, SSL and AE's performance drop less under distribution shift**: The bottom row of fig. 2 compares the shift sensitivity ($s$) of different models (see section 2 for definition). Smaller $s$ is desirable as it indicates lower sensitivity to distribution shift. Results show that SSL and AE algorithms are significantly more stable under distribution shift than SL;
- **Generalisation performance on distribution shift tasks can be significantly improved by re-training the linear head:** We show a large performance boost for all models, when evaluated using linear head trained on a small amount of OOD data, in contrast to the baseline linear head trained on ID data. The surprising gain of this cheap procedure, even on realistic problems, highlights the importance of isolating the linear head bias when evaluating generalisation performance.

## 2 SETING UP

In section 1 we identified three problems in the existing literature that we wish to address in this work. In this section, we will introduce the necessary experimental set-up in further details. In brief, we compare eight ML algorithms on six datasets using three relevant metrics.

**Algorithms:** *×3 SSL, ×4 AE, ×1 SL.* We compare seven unsupervised representation learning algorithms against SL, including three SSL algorithms 1) SimCLR (Chen et al., 2020a), 2) SimSiam (Chen and He, 2021), and 3) BYOL (Grill et al., 2020); and four AE algorithms 1) Autoencoder (Rumelhart et al., 1985), 2) Variational Autoencoder (VAE) (Kingma and Welling, 2014), 3) $\beta$-VAE (Higgins et al., 2017) and 4) Importance Weighted Autoencoder (IWAE) (Burda et al., 2016). These popular methods in SSL and latent generative models have not yet been systematically evaluated under distribution shift tasks prior to our work. We compare the performance of these models against a standard supervised learning (SL) algorithm used as a representation learning model.

**Datasets:** *×2 synthetic, ×2 realistic, ×2 controllable shift.* We evaluate our models on two synthetic datasets, namely MNIST-CIFAR (Shah et al. (2020); see section 3.1.1) and CdSprites (Shi et al. (2022); see section 3.1.2), as well as two realistic datasets from WILDS (Koh et al., 2021): Camelyon17 and FMoW (see section 3.2.1). However, as mentioned in section 1.1, both the synthetic and the realistic datasets have their own drawbacks. To further understand the models' performance and draw conclusions that are generalisable, we also provide a framework for creating *controllable shift* datasets from realistic datasets like those in WILDS, by subsampling the data to introduce spurious correlations between the domain and label information in the training set. Changing this correlation varies the degree of distribution shift between the (ID) train and (OOD) test split, which allows us to analyse the models' performance more effectively under realistic, yet controllable, distribution shift. We refer to this controllable shift versions of the two datasets Camelyon17-CS and FMoW-CS, and provide further details on the datasets in section 3.2.2.

**Evaluation:** *3 metrics.* Before discussing our proposed metrics, we first define some necessary notations. We separate a model trained to perform a classification task into two parts, namely, 1) backbone $f$, denoting the part of the model that generates representations from the data, and 2) final linear head $c$, which takes the representations from $f$ and outputs the final prediction. Further, we refer to the final linear head trained on representations from the ID train set as $c_i$, and that trained on representations from the OOD test set as $c_o$. Since the backbone $f$ is always trained on the ID

train set, we do not make any notation distinction on its training distribution. We also denote the accuracy of $f$ and $c$ on the ID test data as $\text{acc}_i(f, c)$, and on the OOD test data as $\text{acc}_o(f, c)$.

As noted in section 1.1, we report the OOD accuracy of the algorithms using linear heads trained on OOD data (instead of those trained on ID data as per standard practice), i.e. $\text{acc}_o(f, c_o)$. This is necessary to disentangle the bias of the linear head from that of the representations. To highlight the importance of this treatment in isolating the generalisability of the representation learning algorithm from the that of the linear head, we also define the *linear head bias*. It is the difference between the OOD test accuracy evaluated by OOD linear head and that evaluated by the ID linear head, i.e.

$$b = \text{acc}_o(f, c_o) - \text{acc}_o(f, c_i). \tag{1}$$

In a related work, Taori et al. (2020) proposed to evaluate the *effective robustness* of OOD generalisation defined as $\rho = \text{acc}_i(f, c_i) - \text{acc}_o(f, c_i)$, which quantifies the drop in performance (e.g. accuracy) when evaluating the model on OOD test set vs. ID test set. A small $\rho$ is desirable, as it indicates that the performance of the model is relatively insensitive to a distribution shift[2]. However, we note that a simple decomposition of *effective robustness* ($\rho$) shows a hidden *linear head bias* ($b$) term

$$\underbrace{\text{acc}_i(f, c_i) - \text{acc}_o(f, c_i)}_{\text{effective robustness } \rho} = \text{acc}_i(f, c_i) - \text{acc}_o(f, c_i) - \text{acc}_o(f, c_o) + \text{acc}_o(f, c_o)$$

$$= \underbrace{\text{acc}_o(f, c_o) - \text{acc}_o(f, c_i)}_{\text{linear head bias } b} + \underbrace{\text{acc}_i(f, c_i) - \text{acc}_o(f, c_o)}_{\text{shift sensitivity } s}. \tag{2}$$

Thus, we remove the effect of the linear head bias by subtracting $b$ from $\rho$ and reporting the last term in eq. (2). We refer to this as *shift sensitivity*: $s = \rho - b$. Alternatively, it is the difference between the OOD accuracy using linear head trained on OOD data, and ID accuracy using linear head trained on ID data. Larger $s$ marks higher sensitivity of $f$ to distribution shift, which is, possibly, dangerous for the deployment of such models. In summary, for each experiment we report the following three metrics: OOD linear head accuracy $\text{acc}_o(f, c_o)$, linear head bias $b$ and shift sensitivity $s$.

## 3    EXPERIMENTAL RESULTS

We perform a hyperparameter search on learning rate, scheduler, optimiser, representation size, etc. for each model. We use the standard SSL augmentations proposed in He et al. (2020); Chen et al. (2020b) for all models to ensure a fair comparison. See appendix B for details.

### 3.1    SYNTHETIC DISTRIBUTION SHIFT

In this section, we evaluate the performance of SL, SSL and AE algorithms on synthetic distribution shift tasks, utilising the MNIST-CIFAR dataset (Shah et al., 2020) and the CdSprites dataset (Shi et al., 2022). All results are averaged over 5 random seeds.

#### 3.1.1    MNIST-CIFAR

*Finding: Under this extreme distribution shift setting, SSL and AE significantly outperform SL. The OOD accuracy of SSL and AE can be notably improved by retraining the linear head on OOD data, however the OOD accuracy of SL remains low even with the OOD-trained linear head.*

The MNIST-CIFAR dataset consists of concatenations of images from two classes of MNIST and CIFAR-10. In each concatenated image, the classes of the two datasets are either correlated or uncorrelated depending on the split as discussed below (See fig. 1, *MNIST-CIFAR* for an example):

- **ID train, test**: Correlation between MNIST and CIFAR-10 labels is one. Each image belongs to one of the two classes: 1) MNIST "0" and CIFAR-10 "`automobile`", and 2) MNIST "1" and CIFAR-10 "`plane`" (Figure 1, *top row*);
- **OOD train, test**: Zero correlation between MNIST and CIFAR-10 labels, images from the two classes are randomly paired (Figure 1, *bottom row*).

Since the MNIST features are much simpler than the CIFAR features, a model trained on the ID train set can use MNIST only to predict the label, even though the CIFAR images are just as predictive

---

[2]We negate the original definition of effective robustness from Taori et al. (2020) for ease of understanding.

Table 1: Evaluations on the MNIST-CIFAR dataset. We report accuracy on MNIST and CIFAR trained using OOD linear head ($\text{acc}_o(f, c_o)$), linear head bias ($b$) and shift sensitivity ($s$).

| Regime | Method | MNIST (%) | | | CIFAR (%) | | |
|---|---|---|---|---|---|---|---|
| | | $\text{acc}_o(f, c_o) \uparrow$ | $s \downarrow$ | $b$ | $\text{acc}_o(f, c_o) \uparrow$ | $s \downarrow$ | $b$ |
| AE | AE | 99.9 ($\pm 1e$-$2$) | 0.0 ($\pm 1e$-$2$) | 0.0 ($\pm 2e$-$3$) | 81.1 ($\pm 1e$+$0$) | 18.8 ($\pm 1e$+$0$) | 30.2 ($\pm 1e$+$0$) |
| | VAE | 99.8 ($\pm 8e$-$3$) | -0.1 ($\pm 9e$-$3$) | 0.5 ($\pm 1e$-$4$) | 79.7 ($\pm 4e$+$0$) | 20.2 ($\pm 3e$+$0$) | 29.2 ($\pm 6e$+$0$) |
| | IWAE | 99.8 ($\pm 9e$-$3$) | 0.0 ($\pm 4e$-$3$) | 0.1 ($\pm 5e$-$3$) | 80.8 ($\pm 2e$+$0$) | 19.0 ($\pm 3e$+$0$) | 30.0 ($\pm 4e$+$0$) |
| | $\beta$-VAE | 99.8 ($\pm 2e$-$2$) | 0.0 ($\pm 4e$-$2$) | -0.1 ($\pm 3e$-$2$) | 78.0 ($\pm 3e$+$0$) | 21.8 ($\pm 4e$+$0$) | 28.0 ($\pm 4e$+$0$) |
| *AE average* | | *99.8 ($\pm 1e$-$2$)* | *0.0 ($\pm 1e$-$2$)* | *0.1 ($\pm 9e$-$3$)* | *79.9 ($\pm 3e$+$0$)* | *20.0 ($\pm 4e$+$0$)* | *29.3 ($\pm 4e$+$0$)* |
| SSL | SimCLR | 99.7 ($\pm 1e$-$2$) | 0.2 ($\pm 1e$-$3$) | -0.2 ($\pm 3e$-$3$) | 85.8 ($\pm 1e$+$0$) | 14.1 ($\pm 2e$+$0$) | 35.5 ($\pm 1e$+$0$) |
| | SimSiam | 99.8 ($\pm 2e$-$1$) | 0.1 ($\pm 2e$-$1$) | 0.0 ($\pm 9e$-$2$) | 87.8 ($\pm 2e$+$0$) | 12.1 ($\pm 2e$+$0$) | 35.6 ($\pm 4e$+$0$) |
| | BYOL | 99.8 ($\pm 4e$-$2$) | 0.0 ($\pm 1e$-$2$) | 0.9 ($\pm 8e$-$3$) | 84.8 ($\pm 9e$-$1$) | 15.0 ($\pm 1e$+$0$) | 33.2 ($\pm 1e$+$0$) |
| *SSL average* | | *99.8 ($\pm 8e$-$2$)* | *0.1 ($\pm 5e$-$2$)* | *0.2 ($\pm 3e$-$2$)* | *86.1 ($\pm 2e$+$0$)* | *13.7 ($\pm 2e$+$0$)* | *34.8 ($\pm 4e$+$0$)* |
| SL | Supervised | 97.7 ($\pm 9e$-$1$) | 1.4 ($\pm 1e$+$0$) | -0.3 ($\pm 1e$+$0$) | 51.5 ($\pm 1e$+$0$) | 47.6 ($\pm 1e$+$0$) | 0.8 ($\pm 9e$-$1$) |

(Shah et al., 2020). This results in poor performance when predicting the CIFAR label on the OOD test set, where there is no correlation between the MNIST and CIFAR labels.

We train a CNN backbone on the ID train set using the eight SL, SSL and AE algorithms listed in section 2. At test time, we freeze the backbone and train two linear heads on ID train and OOD train set respectively, and evaluate their performance on the ID and OOD test set to compute 1) OOD linear head accuracy $\text{acc}_o(f, c_o)$, 2) shift sensitivity $s$ and, 3) linear head bias $b$. See results in table 1.

We observe that all models achieve near perfect performance when predicting the MNIST label on OOD test set, all with low shift sensitivity and small linear head bias. However, when predicting the labels of the more complex CIFAR images, unsupervised algorithms have a clear advantage over the supervised one: SSL achieves the highest OOD accuracy at $86.1\%$, followed by AE at $79.9\%$ and SL at $51.5\%$ (near random). The shift sensitivity $s$ of the three objectives follow a similar trend, with SSL and AE scoring significantly lower than SL. This indicates that unsupervised representations are significantly less sensitive to distribution shift compared to those from SL, with the latter suffering a drop as large as $47.6\%$. Interestingly, the classifier head bias $b$ for SSL and AE are relatively high (around $30\%$), and is very low for SL ($0.8\%$), indicating that the representations learned from SL is intrinsically un-robust to distribution shift. That is, while there exist (linearly separable) CIFAR features in the representations of SSL and AE that can be extracted using a linear head trained on un-biased (OOD) data, these features are absent from the representations of SL.

### 3.1.2 CDSPRITES

***Finding:*** *Similar to MNIST-CIFAR, under extreme distribution shift, SSL and AE are better than SL; when the shift is less extreme, SSL and SL achieve comparably strong OOD generalisation performance while AE's performance is much weaker.*

CdSprites is a colored variant of the popular dSprites dataset (Matthey et al., 2017), which consists of images of 2D sprites that are procedurally generated from multiple latent factors. The CdSprites dataset induces a spurious correlation between the color and shape of the sprites, by coloring the sprites conditioned on the shape following a controllable correlation coefficient $r_{\text{id}}$. See fig. 1 for an example: when $r_{\text{id}} = 1$ color is completely dependent on shape (*top row*, oval-purple, heart-cyan, square-white), and when $r_{\text{id}} = 0$, color and shape are randomly matched (*bottom row*).

Shi et al. (2022) observes that when $r_{\text{id}}$ is high, SL model tend to use color only to predict the label while ignoring shape features due to the texture bias of CNN (Geirhos et al., 2019; Brendel and Bethge, 2019). First, we consider the setting of extreme distribution shift similar to MNIST-CIFAR by setting $r_{\text{id}} = 1$ in the ID train and test splits. In the OOD train and test splits, the correlation coefficient is set to zero to investigate how well the model learns both the shape and the color features. Table 2 reports the three metrics of interest using the same evaluation protocol as before.

Similar to MNIST-CIFAR, we observe that all models achieve near perfect performance when predicting the simpler feature, i.e. color on the OOD test set. However, when predicting shape, the more complex feature on the OOD test set, SSL (and also AEs to a lesser extent) is far superior to SL. Additionally, the shift sensitivity of SSL (and AE to a lesser extent) are much smaller than SL, indicating that SSL/AE models are more robust to extreme distribution shift. The linear head bias also follows a similar trend as for MNIST-CIFAR, showing that representations learned using SL methods are inherently not robust to spurious correlations. This is not the case for SSL and AE algorithms where a large linear head bias shows that is the ID linear heads and not the representations that injects the bias.

Table 2: Evaluations on the CdSprites dataset with $r_{\mathrm{id}} = 1.0$. We report accuracy for color and shape classifiers trained using OOD linear head ($\mathrm{acc}_o(f, c_o)$), linear head bias ($b$) and shift sensitivity ($s$).

| Regime | Method | Color classification (%) | | | Shape classification (%) | | |
|---|---|---|---|---|---|---|---|
| | | $\mathrm{acc}_o(f, c_o) \uparrow$ | $s \downarrow$ | $b$ | $\mathrm{acc}_o(f, c_o) \uparrow$ | $s \downarrow$ | $b$ |
| | AE | 100.0 ($\pm$0e+0) | 0.0 ($\pm$2e-3) | 0.3 ($\pm$5e-1) | 46.1 ($\pm$6e-1) | 53.9 ($\pm$6e-1) | 12.7 ($\pm$5e-1) |
| AE | VAE | 99.7 ($\pm$3e-1) | 0.3 ($\pm$3e-1) | -0.3 ($\pm$3e-1) | 52.4 ($\pm$2e+0) | 47.6 ($\pm$2e+0) | 18.9 ($\pm$3e+0) |
| | IWAE | 100.0 ($\pm$0e+0) | 0.0 ($\pm$2e-3) | 0.4 ($\pm$5e-1) | 58.9 ($\pm$2e+0) | 41.1 ($\pm$2e+0) | 25.6 ($\pm$2e+0) |
| *AE average* | | *99.9 ($\pm$9e-1)* | *0.1 ($\pm$9e-2)* | *0.1 ($\pm$4e-1)* | *52.5 ($\pm$2e+0)* | *47.5 ($\pm$2e+0)* | *19.1 ($\pm$2e+0)* |
| | SimCLR | 100.0 ($\pm$0e+0) | 0.0 ($\pm$0e+0) | 0.0 ($\pm$1e-1) | 87.8 ($\pm$5e-1) | 12.2 ($\pm$5e-1) | 54.5 ($\pm$5e-1) |
| SSL | SimSiam | 100.0 ($\pm$0e+0) | 0.0 ($\pm$0e+0) | 0.1 ($\pm$1e-1) | 69.2 ($\pm$2e+0) | 30.8 ($\pm$2e+0) | 35.6 ($\pm$2e+0) |
| | BYOL | 100.0 ($\pm$0e+0) | 0.0 ($\pm$0e+0) | 0.0 ($\pm$0e+0) | 91.1 ($\pm$4e+0) | 8.9 ($\pm$4e+0) | 57.9 ($\pm$4e+0) |
| *SSL average* | | *100.0 ($\pm$0e+0)* | *0.0 ($\pm$0e+0)* | *0.1 ($\pm$3e-2)* | *82.7 ($\pm$2e+0)* | *17.3 ($\pm$2e+0)* | *49.3 ($\pm$4e+0)* |
| SL | Supervised | 100.0 ($\pm$0e+0) | 0.0 ($\pm$0e+0) | 0.0 ($\pm$3e-2) | 44.0 ($\pm$7e-1) | 56.0 ($\pm$7e-1) | 10.7 ($\pm$7e-1) |

**Controllable distribution shift**  We extend this experiment to probe the performance of these algorithms under *varying* degrees of distribution shifts. We generate three versions of the CdSprites dataset with three different correlation coefficients $r_{\mathrm{id}} \in \{0, 0.5, 1\}$ of the ID train set. As before, the correlation coefficient of the OOD split is set to zeroThe rest of the experimental protocol stays the same. The OOD test accuracy and the shift sensitivity for varying $r_{\mathrm{id}}$ is plotted in fig. 3 and a detailed breakdown of results is available in appendix A.

Figure 3 shows that despite increasing distribution shift between the ID and OOD splits (with increasing $r_{\mathrm{id}}$) the OOD performance of SSL and AE does not suffer. However, the OOD accuracy of SL plummets and its shift sensitivity explodes at $r_{\mathrm{id}} = 1$. Interestingly, SSL maintains a high OOD test accuracy regardless of the level of distribution shift: when $r_{\mathrm{id}} < 1$ its performance is on par with SL, and when the distribution shift becomes extreme with $r_{\mathrm{id}} = 1$ it significantly outperforms SL both in terms of accuracy and shift sensitivity. In comparison, AE models' accuracy lingers around $50\%$, with increasingly higher shift sensitivity as $r_{\mathrm{id}}$ increases. However, under extreme distribution shift with $r_{\mathrm{id}} = 1$ it still performs better than SL, with slightly higher OOD accuracy and lower shift sensitivity.

## 3.2 REAL-WORLD DISTRIBUTION SHIFT

In this section we investigate the performance of different objectives on real-world distribution shift tasks. We use two datasets from WILDS (Koh et al., 2021): 1) Camelyon17, which contains tissue scans acquired from different hospitals, and the task is to determine if a given patch contains breast cancer tissue; and 2) FMoW, which features satellite images of landscapes on five different continents, with the classification target as the type of infrastructure. See examples in Figure 1. Following the guidelines from WILDS benchmark, we perform 10 random seed runs for all Camelyon17 experiment and 3 random seed runs for FMoW. The error margin in Figure 3 represent standard deviation.

### 3.2.1 ORIGINAL WILDS DATASETS

***Findings:*** *SL is significantly more sensitive to distribution shift than SSL and AE; representations from SSL obtain higher OOD accuracy than SL on Camelyon17 but lower on FMoW. AE is consistently the least sensitive to distribution shift though it has the lowest accuracy. The performance of all models significantly improves by retraining the linear head on a small amount of OOD data.*

The original Camelyon17 and FMoW dataset from WILDS benchmark both contains the following three splits: ID train, OOD validation and OOD test. We further create five splits specified as follows:

- **ID train, test**: Contains 90% and 10% of the original ID train split, respectively;
- **OOD train, test**: Contains 10% and 90% of the original OOD test split, respectively;
- **OOD validation**: Same as the original OOD validation split.

Following WILDS, we use OOD validation set to perform early stopping and choose hyperparameters; we also use DenseNet-121 (Huang et al., 2017) as the backbone for all models. We follow similar evaluation protocol as previous experiments, and in addition adopt 10-fold cross-validation for the OOD train and test set. See results in Tables 3 and 4, where following WILDS, we report performance on Camelyon17 using standard average accuracy and on FMoW using worst-group accuracy.

One immediate observation is that in contrast to our previous experiments on synthetic datasets, SL's OOD accuracy is much higher in comparison on realistic distribution shift tasks: it is the best performing model on FMoW with 35.6% worst-group accuracy on OOD test set; its OOD accuracy is the lowest on Camelyon17, however it is only 3% worse than the highest accuracy achieved by SSL

Table 3: Evaluations on test set of Camelyon17, all metrics computed using average accuracy.

| Regime | Method | Metrics (%) | | |
|---|---|---|---|---|
| | | $\mathrm{acc}_o(f, c_o) \uparrow$ | $s \downarrow$ | $b$ |
| AE | AE | 84.4 ($\pm 2e{+}0$) | -0.6 ($\pm 1e{+}0$) | 12.7 ($\pm 2e{+}0$) |
| | VAE | 88.1 ($\pm 2e{+}0$) | 0.5 ($\pm 2e{+}0$) | 39.0 ($\pm 2e{+}0$) |
| | IWAE | 88.1 ($\pm 1e{+}0$) | -0.9 ($\pm 3e{+}0$) | 39.1 ($\pm 4e{+}0$) |
| | $\beta$-VAE | 87.1 ($\pm 4e{+}0$) | 0.2 ($\pm 4e{+}0$) | 36.0 ($\pm 5e{+}0$) |
| *AE average* | | *86.9 ($\pm 2e{+}0$)* | *-0.2 ($\pm 3e{+}0$)* | *31.7 ($\pm 3e{+}0$)* |
| SSL | SimCLR | 92.7 ($\pm 2e{+}0$) | 0.4 ($\pm 1e{+}0$) | 8.3 ($\pm 1e{+}0$) |
| | SimSiam | 86.7 ($\pm 1e{+}0$) | 3.1 ($\pm 1e{+}0$) | 7.9 ($\pm 3e{+}0$) |
| | BYOL | 89.9 ($\pm 1e{+}0$) | 1.4 ($\pm 1e{+}0$) | 10.3 ($\pm 2e{+}0$) |
| *SSL average* | | *89.8 ($\pm 1e{+}0$)* | *1.6 ($\pm 1e{+}0$)* | *8.8 ($\pm 2e{+}0$)* |
| SL | Supervised | 86.8 ($\pm 2e{+}0$) | 3.5 ($\pm 1e{+}0$) | 7.4 ($\pm 3e{+}0$) |

Table 4: Evaluations on test set of FMoW, all metrics computed using worst-group accuracy.

| Regime | Method | Metrics (%) | | |
|---|---|---|---|---|
| | | $\mathrm{acc}_o(f, c_o) \uparrow$ | $s \downarrow$ | $b$ |
| AE | AE | 26.9 ($\pm 9e{-}3$) | 6.4 ($\pm 6e{-}3$) | 5.8 ($\pm 1e{-}2$) |
| | VAE | 21.7 ($\pm 6e{-}3$) | 4.7 ($\pm 4e{-}3$) | 8.0 ($\pm 2e{-}2$) |
| | IWAE | 20.9 ($\pm 2e{-}2$) | 5.5 ($\pm 1e{-}2$) | 7.8 ($\pm 1e{-}2$) |
| | $\beta$-VAE | 21.7 ($\pm 5e{-}3$) | 3.4 ($\pm 6e{-}3$) | 7.6 ($\pm 8e{-}3$) |
| *AE average* | | *22.8 ($\pm 3e{-}2$)* | *5.0 ($\pm 7e{-}3$)* | *7.3 ($\pm 1e{-}2$)* |
| SSL | SimCLR | 29.9 ($\pm 6e{-}3$) | 10.7 ($\pm 6e{-}3$) | 7.6 ($\pm 7e{-}3$) |
| | SimSiam | 27.8 ($\pm 2e{-}2$) | 4.6 ($\pm 1e{-}2$) | 6.3 ($\pm 2e{-}2$) |
| | BYOL | 31.3 ($\pm 1e{-}2$) | 12.1 ($\pm 7e{-}3$) | 7.9 ($\pm 1e{-}2$) |
| *SSL average* | | *29.6 ($\pm 2e{-}2$)* | *9.1 ($\pm 9e{-}3$)* | *7.3 ($\pm 1e{-}2$)* |
| SL | Supervised | **35.6** ($\pm 7e{-}3$) | 37.7 ($\pm 4e{-}2$) | 6.9 ($\pm 9e{-}3$) |

(89.8%). This highlights the need to study realistic datasets along with synthetic ones. Nonetheless, we find that SSL is still the best performing method on Camelyon17 and achieves competitive performance on FMoW with accuracy 29.6% — despite learning without labels! AE has much lower OOD accuracy on FMoW compared to the other two methods: we believe this is due to its reconstruction-based objective wasting modelling capacity on high frequency details, a phenomenon frequently observed in prior work (Bao et al., 2021; Ramesh et al., 2021). Note that the standard deviation for all three methods are quite high for Camelyon17: this is a known property of the dataset and similar pattern is observed across most methods on WILDS benchmark (Koh et al., 2021).

In terms of shift sensitivity, unsupervised objectives including SSL and AE consistently outperforms SL — this stands out the most on FMoW, where the shift sensitivity of SSL and AE are 9.1% and 5.0% respectively, while SL is as high as 37.7%. This observation further validates our previous finding on synthetic datasets, that SSL and AE's ID accuracy is a relatively reliable indication of their generalisation performance, while SL can undergo a huge performance drop under distribution shift, which can be dangerous for the deployment of such models. We highlight that, in sensitive application domains, a low shift sensitivity is an important criterion as it implies that the model's performance will remain consistent when the distribution shifts. Another interesting observation here is that for all objectives on both datasets, the classifier bias $b$ is consistently high. This indicates the bias of the linear classification head plays a significant role even for real world distribution shifts, and that it is possible to mitigate this effect by training the linear head using a small amount of OOD data (in this case 10% of the original OOD test set).

### 3.2.2 WILDS DATASETS WITH CONTROLLABLE SHIFT

***Findings:*** *SL's OOD accuracy drops as more the distribution shift becomes more challenging, with SSL being the best performing model when the distribution shift is the most extreme. The shift sensitivity of SSL and AE are consistently lower than SL regardless of the level of shift.*

To examine models' generalisation performance under different levels of distribution shift, we create versions of these realistic datasets with *controllable shifts*, which we name Camelyon17-CS and FMoW-CS. Specifically, we subsample the ID train set of these datasets to artificially create spurious correlation between the domain and label. For instance, given dataset with domain A, B and label 0, 1, to create a version of the dataset where the spurious correlation is 1 we would sample only examples with label 0 from domain A and label 1 from domain B. See Appendix C for further details.

Similar to CdSprites, we create three versions of both of these datasets with the spurious correlation coefficient $r_{id} \in \{0, 0.5, 1\}$ in ID (train and test) sets. The OOD train, test and validation set remains unchanged[3]. Using identical experimental setup as in section 3.2.1, we report the results for Camelyon17-CS and FMoW-CS in Figure 3 with detailed numerical results in Appendix A.2.

For both datasets, the OOD test accuracy of all models drop as the spurious correlation $r_{id}$ increases (*top row* in Figure 3). However, this drop is the far more obvious in SL than in SSL and AE: when $r_{id} = 1$, SL's accuracy is 10% lower than SSL on Camelyon17 and 2% lower on FMoW — a significant drop from its original 3% lag on Camelyon17 and 5% lead on FMoW. This demonstrates that SL is less capable of dealing with more challenging distribution shift settings compared to SSL and AE. In terms of shift sensitivity (*bottom row,* fig. 3), SL's remains the highest regardless of $r_{id}$; curiously, we see a decrease in SL's shift sensitivity as $r_{id}$ increases in FMoW-CS, however this has more to do with the ID test set accuracy decreasing due to the subsampling of the dataset.

---

[3]Note that even when $r_{id} = 0$, distribution shift between the ID and OOD splits exists, as the spurious correlation is not the only source of the distribution shift.

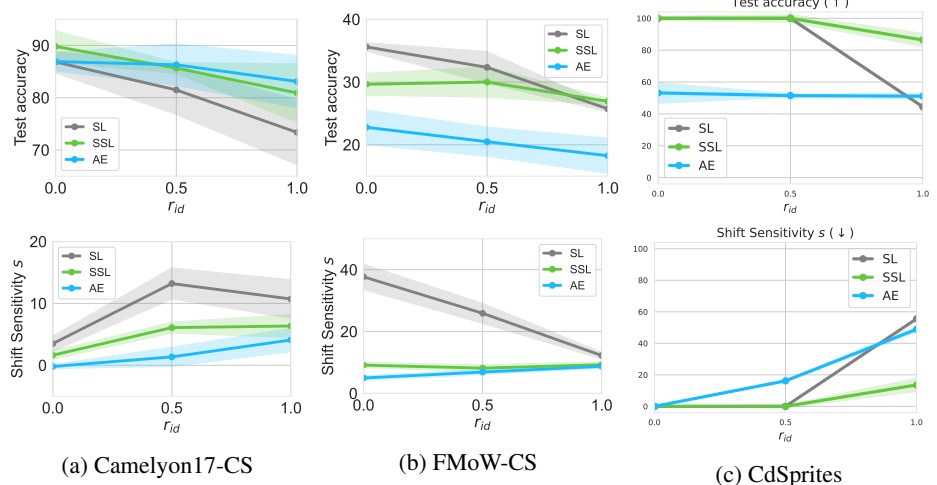

Figure 3: Evaluations on Camelyon17-CS, FMoW-CS, and CdSprites with $r_{\mathrm{id}} \in \{0, 0.5, 1.0\}$. We report OOD test accuracy using OOD-trained linear head ($\mathrm{acc}_o(f, c_o)$) and shift sensitivity ($s$). Blue lines are results averaged over AE models, green lines are SSL models and grey is SL.

## 4 RELATED WORK

While we are the first to systematically evaluate the OOD generalisation performance of unsupervised learning algorithms, there are other insightful work that considers the robustness to distribution shift of other existing, non-specialised methods/techniques. For instance, Liu et al. (2022) studies the impact of different pre-training set-ups to distribution shift robustness, including dataset, objective and data augmentation. Ghosal et al. (2022) focuses on architecture, and found that Vision Transformers are more robust to spurious correlations than ConvNets when using larger models and are given more training data; further, Liu et al. (2021) found that SSL is more robust to data imbalance. Azizi et al. (2022) also performed extensive studies on the generalisation performance of SSL algorithms on medical data. Interestingly, Robinson et al. (2021) also investigates the robustness of contrastive-SSL methods against extreme spurious correlation (i.e.simplicity bias). However, their work did not consider the linear head bias found in (Kirichenko et al., 2022; Kang et al., 2020) and led to opposing conclusions. In contrast, our work investigates the distribution shift performance of unsupervised algorithms, with experiments on both synthetic and realistic settings that go beyond the data imbalance regime. By isolating the linear head bias in our experiments, we find that unsupervised, especially SSL-learned representations, achieves similar if not better generalisation performance than SL under a wide range of distribution shift settings. See Appendix D for a more detailed discussion on distribution shift problems.

## 5 CONCLUSION AND FUTURE WORK

In this paper, we investigate the robustness of both unsupervised (AE, SSL) and supervised (SL) objectives for distribution shift. Through extensive and principled experiments on both synthetic and realistic distribution shift tasks, we find unsupervised representation learning algorithms to consistently outperform SL when the distribution shift is extreme. In addition, we see that SSL's OOD accuracy is comparable, if not better to SL in all experiments. This is particularly crucial, as most work studying distribution shift for images are developed in the SL regime. We hope that these results inspire more future work on unsupervised/semi-supervised representation learning methods for OOD generalisation. Another important finding is that unsupervised models' performance remains relatively stable under distribution shift. This is especially crucial for the real-world application of these machine learning systems, as this indicates that the ID performance of SSL/AE algorithms are a more reliable indicator of how they would perform in different environments at deployment, while that of SL is not. It is also worth noting that while models trained with AE objectives are consistently the least sensitive to distribution shift on realistic datasets, their OOD performance can be low especially when presented with complex data (such as FMoW). This is consistent to the observation in prior work that these models can waste modelling capacity on high frequency details, and suggests that one should be careful about employing AE algorithms on large scale, complex tasks. Finally, a key contribution of this work is establishing the existence of linear head bias even for realistic distribution shift problems. We believe that using an OOD-trained linear head is necessary to be able to make comparisons between various algorithms irrespective of the final downstream task, and on the other hand, more efforts in the field of distribution shift could be devoted into re-balancing the linear layer.

ACKNOWLEDGEMENTS

YS and PHST were supported by the UKRI grant: Turing AI Fellowship EP/W002981/1 and EPSRC/MURI grant: EP/N019474/1. We would also like to thank the Royal Academy of Engineering and FiveAI. YS was additionally supported by Remarkdip through their PhD Scholarship Programme. ID was supported by the SNSF grant #200021_188466. AS was partially supported by the ETH AI Center postdoctoral fellowship. Special thanks to Alain Ryser for suggesting the design of controllable versions of the WILDS dataset, and to Josh Dillon for helpful suggestions in the early stage of this project.

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

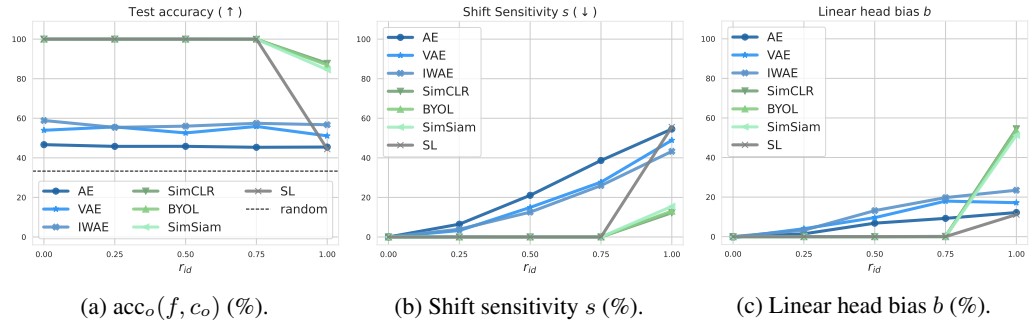

(a) $\mathrm{acc}_o(f, c_o)$ (%).     (b) Shift sensitivity $s$ (%).     (c) Linear head bias $b$ (%).

Figure 4: Evaluations on the CdSprites dataset with $r_{\mathrm{id}} \in \{0.25, 0.5, 0.75, 1.0\}$. We report shape classification accuracy using OOD-trained linear head ($\mathrm{acc}_o(f, c_o)$), shift sensitivity $s$, and linear head bias $b$. Results are shown for *individual* models from the class of AE (blue), SSL (green), and SL (grey) algorithms. The black horizontal line denotes the random baseline (33.3% for three classes).

## A  ADDITIONAL EXPERIMENTAL RESULTS

### A.1  CDSPRITES

In addition to our results in table 2, where we use a dataset with perfectly correlated features ($r_{\mathrm{id}} = 1$) to train the backbones, in fig. 4 we vary $r_{\mathrm{id}}$ to analyse the effect of imperfectly correlated features. Notably, with imperfect correlation ($r_{\mathrm{id}} < 1$), the OOD linear heads trained on top of the SL and SSL backbones perform perfectly. For the AE, we observe that the performance of the OOD linear head does not depend on the correlation $r_{\mathrm{id}}$ in the data used to train the backbones. Our results suggest that with imperfect correlation between features, SL and SSL models learn a linearly separable representation of the features, whereas AE does not.

In fig. 5 we provide an ablation where we also vary $r_{\mathrm{ood}}$, the correlation in the data used to train and evaluate the linear head. Figures 5b and 5c corroborate our results that SSL performs on par with SL for $r_{\mathrm{id}} < 1$ and strictly better when $r_{\mathrm{id}} = 1$. For the AE (Figure 5a), we observe an interesting pattern where the performance of the OOD linear head depends on the OOD correlation $r_{\mathrm{ood}}$, but not on the correlation $r_{\mathrm{id}}$ in the data used to train the backbones. Hence, the ablation corroborates our result that SL and SSL models learn a linearly separable representation of the shape and color features when there is an imperfect correlation between the features, whereas AE does not.

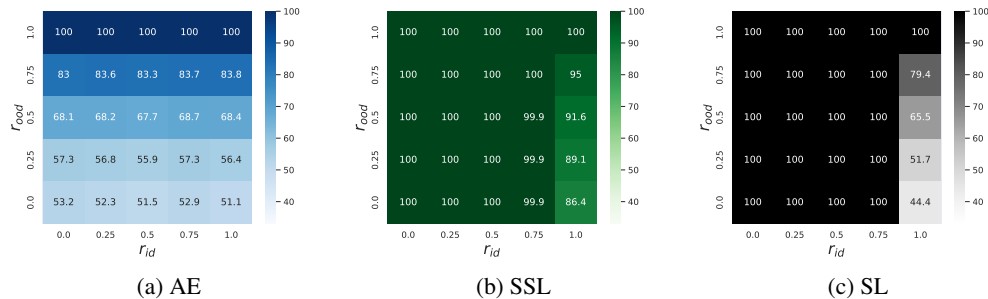

(a) AE      (b) SSL      (c) SL

Figure 5: Correlation coefficient ablation for CdSprites. Shape classification accuracy for the CdSprites experiment with varying correlation of the ID training data ($r_{\mathrm{id}}$, x-axis) and OOD training and test data ($r_{\mathrm{ood}}$, y-axis). Backbones were trained on data with correlation $r_{\mathrm{id}}$ and linear classifiers trained and evaluated on top of the frozen backbones with correlation $r_{\mathrm{ood}}$.

### A.2  CAMELYON17-C AND FMOW-C

In this section we report the numerical results for Camelyon17-C and FMoW-C with $r_{\mathrm{id}} = 0.5$ (see tables 5 and 6) and $r_{\mathrm{id}} = 1$ (see tables 7 and 8).

Table 5: Evaluations on test set of Camelyon17-C with $r_{id} = 0.5$, all metrics computed using average accuracy.

| Regime | Method | Metrics (%) | | |
|---|---|---|---|---|
| | | $acc_o(f, c_o) \uparrow$ | $s \downarrow$ | $b$ |
| AE | AE | 80.4 (±3e+0) | 6.0 (±2e+0) | 19.0 (±4e+0) |
| | VAE | 88.6 (±2e+0) | -0.5 (±1e+0) | 17.8 (±6e+0) |
| | IWAE | 87.8 (±1e+0) | -0.2 (±1e+0) | 26.4 (±6e+0) |
| | $\beta$-VAE | 88.5 (±2e+0) | 0.1 (±9e-1) | 19.7 (±6e+0) |
| *AE average* | | *86.3 (±2e+0)* | *1.4 (±1e+0)* | *20.7 (±5e+0)* |
| SSL | SimCLR | 84.5 (±2e+0) | 8.0 (±1e+0) | 6.6 (±2e+0) |
| | SimSiam | 86.1 (±2e+0) | 5.7 (±1e+0) | 8.3 (±4e+0) |
| | BYOL | 86.4 (±2e+0) | 4.5 (±2e+0) | 8.8 (±4e+0) |
| *SSL average* | | *85.7 (±2e+0)* | *6.1 (±2e+0)* | *7.9 (±3e+0)* |
| SL | Supervised | 81.5 (±5e+0) | 13.2 (±3e+0) | 3.4 (±4e+0) |

Table 6: Evaluations on test set of FMoW-C with $r_{id} = 0.5$, all metrics computed using worst-group accuracy.

| Regime | Method | Metrics (%) | | |
|---|---|---|---|---|
| | | $acc_o(f, c_o) \uparrow$ | $s \downarrow$ | $b$ |
| AE | AE | 23.4 (±1e+0) | 8.6 (±6e-1) | 4.2 (±8e-1) |
| | VAE | 18.7 (±1e+0) | 7.7 (±6e-1) | 1.5 (±6e-1) |
| | IWAE | 18.5 (±2e+0) | 7.3 (±2e+0) | 2.2 (±1e+0) |
| | $\beta$-VAE | 21.4 (±3e-1) | 4.0 (±4e-1) | 3.9 (±7e-1) |
| *AE average* | | *20.5 (±2e+0)* | *6.9 (±8e-1)* | *3.0 (±9e-1)* |
| SSL | SimCLR | 29.5 (±9e-1) | 9.2 (±6e-1) | 6.9 (±7e-1) |
| | SimSiam | 27.9 (±2e+0) | 7.5 (±1e+0) | 4.6 (±1e+0) |
| | BYOL | 32.6 (±3e+0) | 7.9 (±2e+0) | 8.5 (±2e+0) |
| *SSL average* | | *30.0 (±2e+0)* | *8.2 (±1e+0)* | *6.7 (±1e+0)* |
| SL | Supervised | 32.3 (±3e+0) | 25.1 (±3e+0) | 6.0 (±2e+0) |

Table 7: Evaluations on test set of Camelyon17-C with $r_{id} = 1$, all metrics computed using average accuracy.

| Regime | Method | Metrics (%) | | |
|---|---|---|---|---|
| | | $acc_o(f, c_o) \uparrow$ | $s \downarrow$ | $b$ |
| AE | AE | 75.7 (±5e+0) | 7.3 (±2e+0) | 35.1 (±4e+0) |
| | VAE | 86.0 (±3e+0) | 2.7 (±1e+0) | 12.4 (±4e+0) |
| | IWAE | 86.1 (±1e+0) | 2.6 (±7e-1) | 9.1 (±3e+0) |
| | $\beta$-VAE | 84.7 (±2e+0) | 3.8 (±1e+0) | 15.5 (±4e+0) |
| *AE average* | | *83.1 (±3e+0)* | *4.1 (±1e+0)* | *18.0 (±4e+0)* |
| SSL | SimCLR | 85.8 (±8e+1) | 2.8 (±4e-1) | 6.2 (±2e+0) |
| | SimSiam | 82.1 (±1e+0) | 6.0 (±7e-1) | 8.3 (±4e+0) |
| | BYOL | 74.8 (±5e+0) | 10.3 (±2e+0) | -2.2 (±4e+0) |
| *SSL average* | | *80.9 (±2e+0)* | *6.3 (±1e+0)* | *4.1 (±3e+0)* |
| SL | Supervised | 73.4 (±6e+0) | 10.7 (±3e+0) | 5.9 (±8e+0) |

Table 8: Evaluations on test set of FMoW-C with $r_{id} = 1$, all metrics computed using worst-group accuracy.

| Regime | Method | Metrics (%) | | |
|---|---|---|---|---|
| | | $acc_o(f, c_o) \uparrow$ | $s \downarrow$ | $b$ |
| AE | AE | 22.4 (±1e+0) | 10.0 (±6e-1) | 3.8 (±7e-1) |
| | VAE | 16.6 (±9e-1) | 8.6 (±1e+0) | 2.8 (±8e-1) |
| | IWAE | 17.2 (±5e-1) | 8.7 (±6e-1) | 3.8 (±5e-1) |
| | $\beta$-VAE | 16.7 (±3e-1) | 7.9 (±4e-1) | 3.2 (±4e-1) |
| *AE average* | | *18.3 (±3e+0)* | *8.8 (±7e-1)* | *3.4 (±6e-1)* |
| SSL | SimCLR | 26.3 (±1e+0) | 10.6 (±8e-1) | 7.2 (±1e+0) |
| | SimSiam | 27.1 (±5e-1) | 6.3 (±7e-1) | 7.8 (±3e-1) |
| | BYOL | 27.4 (±2e+0) | 11.1 (±1e+0) | 6.5 (±2e+0) |
| *SSL average* | | *26.9 (±6e-1)* | *9.3 (±1e+0)* | *7.2 (±1e+0)* |
| SL | Supervised | 25.7 (±5e-1) | 12.3 (±2e+0) | 5.3 (±1e+0) |

# B  ARCHITECTURE AND HYPERPARAMETERS

In this appendix we list the architecture and hyperparameters used in our experiments. Our code is developed on the amazing `solo-learn` code base (da Costa et al., 2022), which is originally developed as a library for SSL algorithms. For all experiments we follow the standard set of augmentations established in He et al. (2020); Chen et al. (2020b), including random resize crop, random color jittering, random grayscale, random Gaussian blur, random solorisation and random horizontal flip. An exception is the CdSprites experiment where we remove color jittering, as color classification is one of the tasks we are interested in and color jittering would add noise to the labels. For MNIST-CIFAR, we independently apply random augmentation to MNIST and CIFAR respectively (drawn from the same set of augmentations as detailed above) and then concatenate them to construct training examples.

Please see implementation details for each dataset in the respective subsection.

## B.1  MNIST-CIFAR

We use the same hyperparameter search range for models in each category of AE, SSL and SL, as outlined in table 9. The chosen hyperparameters for each model are specified in table 10.

In Shah et al. (2020) where MNIST-CIFAR was originally proposed, authors utilised more complex backbone architecture such as DenseNet and MobileNet. However in our experiments, we find that a lightweight 4-layer CNN can already achieve very high accuracy on both MNIST and CIFAR. The architecture of the CNN we use can be found in table 11. Note that for SL and SSL we only use the encoder and for AE we use the decoder as well. The size of base channel $C$ and latent dimension $L$ are found through hyperparameter search.

Table 9: Hyperparameter search range for MNIST-CIFAR, including base channel size of CNN ($C$), learning rate ($lr.$), weight decay ($wd.$), optimiser ($optim.$), learning rate scheduler ($lr.\ scheduler$).

|  | $C$ | lr. | wd. | Optim. | lr. scheduler |
|---|---|---|---|---|---|
| AE | {16, 32, 64, 128} | {1e-4, 5e-4, 1e-3, 5e-3, 1e-2} | {0, 1e-4} | {Adam, SGD} | {warmup cosine, step, none} |
| SSL | {16, 32, 64, 128} | uniformly sampled from [0.1, 1] | {0, 1e-4} | {Adam, SGD} | {warmup cosine, step, none} |
| SL | {16, 32, 64, 128} | {1e-4, 5e-4, 1e-3, 5e-3, 1e-2, 1e-1, 5e-1} | {0, 1e-4} | {Adam, SGD} | {warmup cosine, step, none} |

Table 10: Chosen hyperparameters for MNIST-CIFAR including latent dimension ($L$), base feature size of CNN ($C$), batch size ($B$), learning rate ($lr.$), weight decay ($wd.$), optimiser ($optim.$), learning rate scheduler ($lr.scheduler$).

|  | $L$ | $C$ | $B$ | lr. | wd. | Optim. | lr. scheduler |
|---|---|---|---|---|---|---|---|
| AE | 128 | 16 | 128 | 1e-3 | 0 | Adam | warmup cosine |
| VAE | 128 | 32 | 128 | 1e-4 | 0 | Adam | warmup cosine |
| IWAE | 128 | 32 | 128 | 1e-4 | 0 | Adam | step |
| $\beta$-VAE | 128 | 16 | 128 | 1e-4 | 0 | Adam | step |
| SimCLR | 128 | 32 | 128 | 6e-1 | 1e-4 | SGD | warmup cosine |
| BYOL | 128 | 64 | 128 | 7e-1 | 0 | SGD | warmup cosine |
| SimSiam | 128 | 128 | 128 | 6e-1 | 1e-5 | SGD | warmup cosine |
| Supervised | 128 | 16 | 128 | 1e-4 | 0 | SGD | warmup cosine |

**Encoder**

Input $\in \mathbb{R}^{3\times64\times32}$
4x4 conv. $C$ stride 2x2 pad 1x1 & ReLU
4x4 conv. $2C$ stride 2x2 pad 1x1 & ReLU
4x4 conv. $4C$ stride 2x2 pad 1x1 & ReLU
4x1 conv. $4C$ stride 2x1 pad 1x0 & ReLU
4x4 conv. $L$ stride 1 pad 0, 4x4 conv. $L$ stride 1x1 pad 0x0

**Decoder**

Input $\in \mathbb{R}^{L}$
4x4 upconv. $4C$ stride 1x1 pad 0x0 & ReLU
4x1 upconv. $4C$ stride 2x1 pad 1x0 & ReLU
4x4 upconv. $2C$ stride 2x2 pad 1x1 & ReLU
4x4 upconv. $C$ stride 2x2 pad 1x1 & ReLU
4x4 upconv. 3 stride 2x2 pad 1x1 & Sigmoid

Table 11: CNN architecture, MNIST-CIFAR dataset.

## B.2 CDSPRITES

We found all models to be relatively robust to hyperparameters, as most configurations result in close to perfect shape and color classification accuracy on the ID validation set. The chosen hyperparameters for each model are specified in table 12. We omit $\beta$-VAE from the comparison, as we empirically found that $\beta = 1$ leads to the best performance on the ID validation set and therefore the results for the $\beta$-VAE would be similar to the VAE. We use the same augmentations (random crops and horizontal flips) for all models and use no color augmentations in order to keep the invariance of the learned representations with respect to color. The encoder and decoder architectures are described in table 13.

Table 14: Hyperparameter search range for Camelyon17, including decoder type, latent dimension ($L$), learning rate (*lr.*), weight decay (*wd.*), optimiser (*optim.*), learning rate scheduler (*lr. scheduler*).

| | Decoder type | $L$ | lr. | wd. | Optim. | lr. scheduler |
|---|---|---|---|---|---|---|
| AE | [CNN, MLP, ResNet] | {256, 512, 1024} | {1e-4, 5e-4, 1e-3, 5e-3, 1e-2} | {0, 1e-4} | {Adam, SGD} | {warmup cosine, step, none} |
| SSL | - | {256, 512, 1024} | {1e-4, 5e-4, 1e-3, 5e-3, 1e-2, 1e-1, 5e-1, 1} | {0, 1e-3, 1e-4, 1e-5} | {Adam, SGD} | {warmup cosine, step, none} |
| SL | - | {256, 512, 1024} | {1e-4, 5e-4, 1e-3, 5e-3, 1e-2, 1e-1, 5e-1} | {0, 1e-4} | {Adam, SGD} | {warmup cosine, step, none} |

Table 12: Chosen hyperparameters for CdSprites including latent dimension ($L$), base feature size of CNN ($C$), batch size ($B$), learning rate (*lr.*), weight decay (*wd.*), optimiser (*optim.*), learning rate scheduler (*lr.scheduler*).

| | L | C | B | lr. | wd. | Optim. | none |
|---|---|---|---|---|---|---|---|
| AE | 512 | 64 | 128 | 5e-5 | 1e-4 | Adam | none |
| VAE | 512 | 64 | 128 | 5e-5 | 1e-4 | Adam | none |
| IWAE | 512 | 64 | 128 | 5e-5 | 1e-4 | Adam | none |
| SimCLR | 64 | 32 | 64 | 5e-3 | 1e-5 | SGD | warmup cosine |
| BYOL | 64 | 32 | 64 | 5e-1 | 1e-5 | SGD | warmup cosine |
| SimSiam | 64 | 32 | 64 | 8e-2 | 1e-5 | SGD | warmup cosine |
| Supervised | 512 | 64 | 128 | 5e-5 | 1e-4 | Adam | none |

**Encoder**

Input $\in \mathbb{R}^{3 \times 64 \times 64}$
4x4 conv. $C$ stride 2x2 pad 1x1 & ReLU
4x4 conv. $2C$ stride 2x2 pad 1x1 & ReLU
4x4 conv. $4C$ stride 2x2 pad 1x1 & ReLU
4x4 conv. $8C$ stride 2x2 pad 1x1 & ReLU
4x4 conv. $L$ stride 1 pad 0

**Decoder**

Input $\in \mathbb{R}^{L}$
4x4 upconv. $8C$ stride 1x1 pad 0x0 & ReLU
4x4 upconv. $4C$ stride 2x2 pad 1x1 & ReLU
4x4 upconv. $2C$ stride 2x2 pad 1x1 & ReLU
4x4 upconv. $C$ stride 2x2 pad 1x1 & ReLU
4x4 upconv. 3 stride 2x2 pad 1x1

Table 13: CNN architecture, CdSprites dataset.

## B.3 CAMELYON17 AND FMoW

For hyperparameters including batch size, max epoch and model selection criteria, we follow the same protocol as in WILDS (Koh et al., 2021): for Camelyon17 we use a batch size of 32, train all models for 10 epochs and select the model that results in the highest accuracy on the validation set, and for FMoW the batch size is 32, max epoch is 60 and model selection criteria is worst group accuracy on OOD validation set. For the rest, we use the same hyperparameter search range for models in each category of AE, SSL and SL, as outlined in table 14. The chosen hyperparameters for Camelyon17 are specified in table 15, and for FMoW in table 16. For Camelyon17-C and FMoW-C we use these same hyperparameters.

Table 15: Chosen hyperparameters for Camelyon17 including latent dimension ($L$), learning rate (*lr.*), weight decay (*wd.*), optimiser (*optim.*), learning rate scheduler (*lr. scheduler*).

| | Decoder | lr. | wd. | Optim. | lr. scheduler |
|---|---|---|---|---|---|
| AE | ResNet | 5e-4 | 1e-5 | SGD | warmup cosine |
| VAE | MLP | 1e-4 | 0 | Adam | none |
| IWAE | MLP | 1e-4 | 0 | Adam | none |
| $\beta$-VAE | MLP | 1e-4 | 0 | Adam | none |
| SimCLR | - | 1e-1 | 0 | SGD | none |
| BYOL | - | 1e-1 | 1e-5 | SGD | warmup cosine |
| SimSiam | - | 1e-1 | 1e-5 | SGD | warmup cosine |
| Supervised | - | 1e-3 | 1e-3 | SGD | none |

Table 16: Chosen hyperparameters for FMoW including latent dimension ($L$), learning rate (*lr.*), weight decay (*wd.*), optimiser (*optim.*), learning rate scheduler (*lr. scheduler*).

| | Decoder | lr. | wd. | Optim. | lr. scheduler |
|---|---|---|---|---|---|
| AE | CNN | 1e-1 | 1e-4 | SGD | none |
| VAE | MLP | 1e-6 | 1e-4 | Adam | step |
| IWAE | MLP | 1e-6 | 1e-4 | Adam | step |
| $\beta$-VAE | MLP | 1e-6 | 1e-4 | Adam | step |
| SimCLR | - | 5e-4 | 1e-3 | SGD | step |
| BYOL | - | 1e-2 | 1e-4 | SGD | step |
| SimSiam | - | 5e-4 | 0 | SGD | step |
| Supervised | - | 1e-4 | 0 | Adam | step |

We follow Koh et al. (2021) and use DenseNet121 (Huang et al., 2017) as backbone architecture. For the decoder of the AE models, we perform hyperparameter search between three architectures: a CNN (see table 17), a simple 3-layer MLP (see table 18) and a ResNet-like decoder with skip connections (see table 19).

| **CNN, Decoder** |
|---|
| Input $\in \mathbb{R}^L$ |
| 4x4 upconv. $8C$ stride 2x2 pad 1x1 & ReLU |
| 4x4 upconv. $8C$ stride 2x2 pad 0x0 & ReLU |
| 4x4 upconv. $4C$ stride 2x2 pad 1x1 & ReLU |
| 4x4 upconv. $2C$ stride 2x2 pad 1x1 & ReLU |
| 4x4 upconv. $C$ stride 2x2 pad 1x1 & ReLU |
| 4x4 upconv. 3 stride 2x2 pad 1x1 & Sigmoid |

Table 17: CNN architecture, Camelyon17 dataset.

| **MLP, Decoder** |
|---|
| Input $\in \mathbb{R}^L$ |
| fc. $2L$ & ReLU |
| fc. $4L$ & ReLU |
| fc. 3*96*96 & ReLU |

Table 18: MLP architecture, Camelyon17 dataset.

| **ResNet, Decoder** |
|---|
| Input $\in \mathbb{R}^L$ |
| fc. 2048 & ReLU |
| 3x3 conv. $16C$ stride 1x1 pad 1x1 |
| 3x3 conv. $16C$ stride 1x1 pad 1x1 |
| x2 upsample |
| 3x3 conv. $8C$ stride 1x1 pad 1x1 |
| 3x3 conv. $8C$ stride 1x1 pad 1x1 |
| x2 upsample |
| 3x3 conv. $8C$ stride 1x1 pad 1x1 |
| 3x3 conv. $8C$ stride 1x1 pad 1x1 |
| 3x3 conv. 3 stride 1x1 pad 1x1 |

Table 19: ResNet decoder architecture, Camelyon17 dataset.

## C CONSTRUCTING CAMELYON17-CS AND FMoW-CS

We subsample Camelyon17 and FMoW dataset to create varying degree of spurious correlation between the domain and label information. We refer to these datasets as Camelyon17-CS and FMoW-CS. To construct such datasets, we first find some domain-label pairing in each dataset, such that if we sample the dataset according to this pairing, the population of each class with respect to the total number of examples in the dataset remains relatively stable. The $r_{id} = 1$ versions of both Camelyon17-CS and FMoW-CS can be acquired by simply subsampling the dataset following the domain label pairing; to ensure fairness in comparison, when constructing the $r_{id} \in \{0, 0.5\}$ versions of these datasets, we first mix in anti-bias samples (i.e. samples that are not in the domain-label

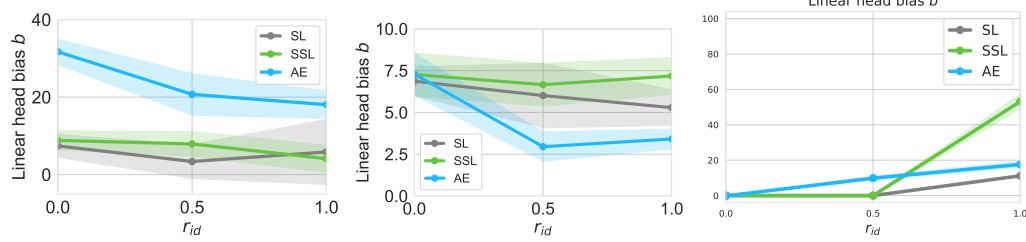

Figure 6: Linear head bias on controllable shift datasets

pairing) to change the spurious correlation, and then subsample the dataset such that the size of the dataset is the same as the $r_{id} = 1$ version.

The domain-label pairing of Camelyon17-CS can be found in table 20 and FMoW-CS in table 21.

**Linear head bias**  We also plot the linear head bias for the experiments conducted on Camelyon17-CS, FMoW-CS, and CdSprites in Figure 6. The experimental protocol follows that from Figure 3.

Table 20: Domain-label pairing for Camelyon17-CS.

| Domain (hospital) | Label |
|---|---|
| Hospital 1, 2 | Benign |
| Hospital 3 | Malignant |

Table 21: Domain-label pairing for FMoW-CS.

| Domain (region) | Label |
|---|---|
| Asia | Military facility, multi-unit residential, tunnel opening, wind farm, toll booth, road bridge, oil or gas facility, helipad, nuclear powerplant, police station, port |
| Europe | Smokestack, barn, waste disposal, hospital, water treatment facility, amusement park, fire station, fountain, construction site, shipyard, solar farm, space facility |
| Africa | Place of worship, crop field, dam, tower, runway, airport, electric substation, flooded road, border checkpoint, prison, archaeological site, factory or powerplant, impoverished settlement, lake or pond |
| Americas | Recreational facility, swimming pool, educational institution, stadium, golf course, office building, interchange, car dealership, railway bridge, storage tank, surface mine, zoo |
| Oceania | Single-unit residential, parking lot or garage, race track, park, ground transportation station, shopping mall, airport terminal, airport hangar, lighthouse, gas station, aquaculture, burial site, debris or rubble |

# D  OTHER RELATED WORK

**Explicit, extreme distribution shift**  Refers to when the features that caused distribution shift is explicit, known, controllable, and in some cases, extreme (e.g. MNIST-CIFAR, CdSprites). This type of settings are popular in works that investigate simplicity bias (Shah et al., 2020), dataset bias (Torralba and Efros, 2011) and shortcut learning (Geirhos et al., 2020; Lapuschkin et al., 2019), as it allows for users to easily adjust the level of distribution shift between train and test set. Various specialised methods that either mitigate or address these problems under the supervised learning regime have been proposed, including Teney et al. (2022) that proposes to find shortcut solutions by ensembles, Luo et al. (2021) that avoids shortcut learning by extracting foreground objects for representation learning only, as well as Torralba and Efros (2011); Kim et al. (2019); Le Bras et al. (2020) that re-sample the dataset to reduce the spurious correlation.

Importantly Kirichenko et al. (2022); Kang et al. (2020) shows that this extreme simplicity bias can be mitigated in some cases by retraining the final linear layer. This is a game changer, as it for the first time decouples the bias of the linear head from that of the main representation learning model. Interestingly, Robinson et al. (2021) also investigates the robustness of contrastive-SSL methods against simplicity bias. However, without training the linear head on OOD data, their finding is opposite to ours — that SSL methods are not able to avoid shortcut solutions.

**Implicit, subtle distribution shift**    This type of problem is commonly seen in realistic distribution shift datasets (such as WILDS) and are often studied in Domain generalisation (DG). In this regime, the training data are sampled from multiple domains, while data from a new, unseen target domain is used as test set. Note that here we omitted the discussion on domain adaptation (DA), as in DA models typically have access to unlabelled data from the target domain, which is different from the settings we consider in this work.

There are mainly two lines of work in DG, namely 1) *Distributional Robustness approaches (DRO)*, which minimises the worst group accuracy to address covariate shift (Gretton et al., 2009a;b) and subpopulation shift (Sagawa et al., 2020; Hu et al., 2018); 2) *Domain invariance*, which consists of methods that directly learn representations that are invariant across domains (Ben-David et al., 2010; Ganin et al., 2016; Wang et al., 2019), encourage the alignment of gradients from different domains (Koyama and Yamaguchi, 2021; Lopez-Paz and Ranzato, 2017; Shi et al., 2022), or optimise for representations that result in the same optimal classifier for different domains (Arjovsky et al., 2019). Apart from these supervised learning methods, the recent advancement of SSL has also inspired works in unsupervised domain generalisation (Zhang et al., 2022; Harary et al., 2022). While all these methods achieved impressive performance, we note that they are all specially designed for DG with the majority of the methods relying on domain information and label information. In contrast, our work studies how existing standard representation learning methods such as SSL and AE performs on DG tasks, with none of the methods relying on human annotations.

