# OpenReview forum: "How robust is unsupervised representation learning to distribution shift?"
_ICLR.cc/2023/Conference — ICLR 2023 poster_

### Official Review · Reviewer_ekJ8 · 2022-10-19

**Confidence:** 4
**Correctness:** 3
**Technical Novelty And Significance:** 2
**Empirical Novelty And Significance:** 2
**Recommendation:** 5

**Clarity, Quality, Novelty And Reproducibility:**

Clarity and Quality: The presented approach or the presentation can be further improved but is reasonable as it is.

Novelty: The novelty side of this paper is weak as most of the work is existing in the literature rather it is astudy on the effectiveness of the existing approach(s)

Reproducibility: The results can be reproduced.

**Strength And Weaknesses:**

# Pros:

- The paper is well-written, easy to understand and follow along.

- The explorations to identify the significance of representations with unsupervised learning in a distribution shift are interesting.

- Literature is covered adequately.

# Cons:


- Significance of the following contributions highlighted is questionable:
  - Systematically evaluate SSL and AE on distribution shift tasks.
  - The proposed solution in the paper is what everyone uses and is a standard practice, how is it different from what already exists or known to the community
  - Controllable-Shift (CS) datasets, are they used only during evaluation or they used in training also?
  - OOD linear head. Again, fine-tuning the pre-trained model on a small set of left out data is also a known practice then what is your contribution here to highlight this as a stand out contribution?

- Figure 2 is referenced first and Figure 2 later, better move Figure 1 to the end of page 3 or page 4.

- Figure 1 also warrants focussed attention of the reader to understand forcing the reader to spend a few extra minutes. Not an ideal presentation style.

- The subsampling used to prepare the CS datasets is not entirely clear. How can you guarantee that the samples picked result in spurious correlations? Are the real world sets by default contains spurious correlations in the entire dataset so that you can pick any subsample of images randomly to mimic this behavior? A small algorithm on CS dataset prep in appendix also could be of value.

- How did you train CNN back bone? No details on hyper params used, network architecture, number of epochs. As per the loss functions used, those eight SL, AE and SSL have the standard ones so that is fine. Also, how long did you fine-tune the pre-trained backbone (f) on the held-out OOD data?

- Another question, why a simple CNN why not the state-of-the-art SSL methods such as contrastive learning or the more recent CNNs architectures such as ResNets, etc.? Will the results hold true with more advanced networks? This question remains more true given the performance of the unsupervised methods on real world datasets, the results in 3.2.1


 ### Minor comments:
 - “This is becaues, …” should be “This is because,...”
 - “under distribution shif, …” should be “under distribution shift, …”

  - Too many forward references, this is not a thesis introduction chapter, but a research paper, please minimize the number of forward references.


**Summary Of The Paper:**

Studying the robustness of learned representations in unsupervised learning such as self-supervised learning or auto encoders as opposed to the traditional supervised learning settings when there exists distribution shifts in data.



**Summary Of The Review:**

Please refer to the strengths and weaknesses. IMO, the weaknesses out-weight the strengths in this paper.

---

> ### Author Response · Authors · 2022-11-08
> **Thank you for your reviews! (1/2)**
>
> We thank the reviewer for suggestions on improving presentations and will make changes in our updated manuscript.
>
> > Q1. Significance of the following contributions highlighted is questionable: "Systematically evaluate SSL and AE on distribution shift tasks." The proposed solution in the paper is what everyone uses and is a standard practice, how is it different from what already exists or known to the community
>
> It is not true that the proposed solution is already standard practice. In fact, on the popular domain generalisation benchmark DomainBed [1], none of the featured 27 SOTA methods adopts unsupervised objectives or uses a re-trained linear head on OOD data for evaluation. We'd be more than happy to include any related works that study or adopt similar methods for distribution shift tasks that the reviewer knows of and would appreciate any such pointers from the reviewer.
>
> > Q2. Controllable-Shift (CS) datasets, are they used only during evaluation or they used in training also?
>
> They are used in training.
>
>
> > Q3. OOD linear head. Again, fine-tuning the pre-trained model on a small set of left out data is also a known practice then what is your contribution here to highlight this as a stand out contribution?
>
> Thank you for your comment! We would like to point out a possible misunderstanding:
>
> 1. Re-training linear head is indeed common, but also necessary in transfer learning setting, as the target data typically features a different task than the source data that the model is trained on; on the other hand, our work studies distribution shift settings where the source and target data feature the same task – re-training the linear classifier on target domain is therefore not necessary, and thus not standard practice, as also noted by reviewers xv8k and bhk8. We show in our experiment that re-training the linear classifier significantly improves generalisation performance even on realistic distribution shift tasks, which is not previously known to the community.
> 2. Prior work [2] and [3] also investigates the performance of SSL on distribution shift tasks. However these work did not take the linear head bias into account, and concluded that representations learned from SSL are worse at generalisation. By re-training the linear head, surprisingly, we came to the opposite conclusion to these prior work. This demonstrates the importance of removing the linear head bias.
>
> Thank you for pointing this out, we will make the distinction between distribution shift vs. transfer learning more obvious in our updated manuscript to highlight our contribution.
>
>
>
>
> > Q4. Figure 2 is referenced first and Figure 2 later, better move Figure 1 to the end of page 3 or page 4. Figure 1 also warrants focussed attention of the reader to understand forcing the reader to spend a few extra minutes. Not an ideal presentation style.
>
> Thank you for your suggestion! We will update our manuscript to improve our presentation.
>
> > Q5. The subsampling used to prepare the CS datasets is not entirely clear. How can you guarantee that the samples picked result in spurious correlations? Are the real world sets by default contains spurious correlations in the entire dataset so that you can pick any subsample of images randomly to mimic this behavior? A small algorithm on CS dataset prep in appendix also could be of value.
>
>
>
> To construct the CS datasets, we subsample the datasets to introduce correlation between domains and labels. The spurious correlation is guaranteed because we have information about the domain and label for each example. For instance, given dataset with domain A, B and label 0, 1, to create a version of the dataset where the spurious correlation is 1 we would sample only examples with label 0 from domain A and label 1 from domain B.
>
>
> We explain this in details in paragraph 1 of Section 3.2.2 as well as Appendix C. In Table 20 and 21 of our manuscript, we included all domain-label pairing used to construct FMoW-CS and Camelyon17-CS, which we hope suffice for future works that are interested in using these datasets.

---

> > ### Author Response · Authors · 2022-11-08
> > **Thank you for your reviews! (2/2)**
> >
> > > Q6. How did you train CNN back bone? No details on hyper params used, network architecture, number of epochs. As per the loss functions used, those eight SL, AE and SSL have the standard ones so that is fine. Also, how long did you fine-tune the pre-trained backbone (f) on the held-out OOD data?
> >
> > All training details, including hyperparameters, network architecture, number of epochs for each dataset are in Appendix B.
> >
> > We did not fine-tune the pre-trained backbone (f) on the held out OOD data.
> >
> > > Q7. Another question, why a simple CNN why not the state-of-the-art SSL methods such as contrastive learning or the more recent CNNs architectures such as ResNets, etc.? Will the results hold true with more advanced networks? This question remains more true given the performance of the unsupervised methods on real world datasets, the results in 3.2.1
> >
> > We did indeed use more advanced networks: for Camelyon17 we use DenseNet-121, and FMoW we use ResNet-50. We only use simple CNN for simpler datasets such as MNIST-CIFAR and CdSprites following practices in the original work [4, 5] where these datasets are first proposed.
> >
> >
> > **References:**
> >
> > [1] Gulrajani, Ishaan, and David Lopez-Paz. “In search of lost domain generalization.” (2020)
> >
> > [2] Joshua Robinson, Li Sun, Ke Yu, Kayhan Batmanghelich, Stefanie Jegelka, and Suvrit Sra. Can contrastive learning avoid shortcut solutions? (2021)
> >
> > [3] Sagawa, Shiori, et al. "Extending the wilds benchmark for unsupervised adaptation." (2021)
> >
> > [4] Harshay Shah, Kaustav Tamuly, Aditi Raghunathan, Prateek Jain, and Praneeth Netrapalli. The pitfalls of simplicity bias in neural networks. (2020)
> >
> > [5] Yuge Shi, Jeffrey Seely, Philip HS Torr, N Siddharth, Awni Hannun, Nicolas Usunier, and Gabriel Synnaeve. Gradient matching for domain generalization. (2022)

---

> ### Author Response · Authors · 2022-11-17
> **Awaiting your reply**
>
> Dear Reviewer,
>
> We believe we have answered all the questions you asked with additional clarifications both in the reply here and in the general comments  as well as updates in the paper. We are eagerly waiting for your response to know if there is something further we could do.

---

> ### Author Response · Authors · 2022-11-27
> **Discussion period coming to an end**
>
> Dear reviewer,
>
> As the discussion period is drawing to an end, we were wondering if you had a chance to look at our response and considered increasing your score for our paper.
>
> Paper authors

---

> ### Comment · Reviewer_ekJ8 · 2022-11-30
> **Thanks for the Response**
>
> After going through the revisions, I am able to raise the score to 5 however, I still believe the innovation side of the paper is weak.

---

### Official Review · Reviewer_JB4R · 2022-10-25

**Confidence:** 3
**Correctness:** 3
**Technical Novelty And Significance:** 3
**Empirical Novelty And Significance:** 3
**Recommendation:** 5

**Clarity, Quality, Novelty And Reproducibility:**

The paper is clearly written. The analysis is relatively sound. I do not find a code repository but appendix includes enough details for reproducibility.

As mentioned, I think the contribution is mainly about conducting rigorous benchmarking research in this domain, which is valuable. The conclusion, though, is relatively known to the community.

**Strength And Weaknesses:**

Strength:

The empirical contribution should be appreciated by the community, as it is something people have been discussing for a long time. It is good to see a concrete evaluation paper.

This paper is very clearly written.

Weakness:

The fact that this paper only discusses image data may limit its impact. To be honest, I think the conclusion is sort of known to the community already. Pretraining in the source domain and fine-tuning the linear layers in the target domain has been shown to be an effective transfer learning method. So I feel what is more interesting in the direction is the large language models or vision-transformers.

On the other hand, I think the elephant in the room is still unknown: To what extent can we really rely on the pretrained representations to do the transfer and adaptation? What is the limit?

The way we set up the problem may also have some impact on the results. For example, the way we "control" the shift in the images may not be generalizable to real-world shifts in, for example, fairness problems/subpopulation shifts, where we cannot easily create shifts in a meaningful way. So, I also suggest adding discussions on the limitations of the work.

In table 4, why the SL accuracy is much higher?



**Summary Of The Paper:**

This paper benchmarks self-supervised learning (SSL) techniques and auto-encoder (AE) techniques under synthetic and realistic distribution shifts. The main conclusion is that the SSL and AE methods are less sensitive to distribution shifts. Also, the linear classification head is the main source of the spurious bias. Finally, the paper also develops ways to create different levels of shifts in real-world shifts datasets.

**Summary Of The Review:**

It will be nice if the scale of the research is a bit larger than the current one. By scale, I mean the coverage of different SSL methods and AE methods, the scale of the dataset, and the architecture. Because what is still unknown to the community is the limit of scaling up in using SSL or AE in domain adaptation/generalization/distribution shifts.

---

> ### Author Response · Authors · 2022-11-08
> **Thank you for your reviews! (1/2)**
>
> We thank the reviewer for the comprehensive reviews and proposing interesting directions for future work in this line of research. We hope the below responses clear up some of the reviewer's concerns.
>
> > Q1. I think the conclusion is sort of known to the community already. Pretraining in the source domain and fine-tuning the linear layers in the target domain has been shown to be an effective transfer learning method.
>
> Thank you for your comment! We would like to address a potential misunderstanding of the setting that we study:
> 1) Re-training linear head is indeed common, but also necessary in a **transfer learning setting**, as the target data typically features a different task than the source data that the model is trained on, hence the linear classifier needs to be "transferred"; on the other hand, our work studies **distribution shift settings** where the source and target data feature the same task -- re-training linear classifier on target domain is therefore not necessary, and thus not at standard practice, as also noted by reviewers xv8k and bhk8. We show in our experiments that re-training the linear classifier significantly improves generalisation performance even on realistic distribution shift tasks, which was not previously known to the community;
> 2) Prior work [1] and [2] also investigates the performance of SSL on distribution shift tasks. However these works did not take the linear head bias into account, and concluded that representations learned from SSL are worse at generalisation. By re-training the linear head, surprisingly, we came to the opposite conclusion to these prior works. This demonstrates the importance of removing the linear head bias when evaluating generalisation performance.
>
> Thank you for pointing this out, we will make the distinction between distribution shift vs. transfer learning more obvious in our updated manuscript to highlight our contribution.
>
>
> > Q2. ...what is more interesting in the direction is the large language models or vision-transformers
>
> How **choice of architectures** affects distribution shift is indeed a very interesting research direction, thank you for your suggestion! In saying that, we believe that our work, which focuses more on how the **choice of learning ojectives** (SSL vs SL vs AE) affects distribution shift, is also of interest to the community. We leave the exploration of architectures and distribution shift to future work.
>
> > Q3.1 The fact that this paper only discusses image data may limit its impact.
>
> > Q3.2 It will be nice if the scale of the research is a bit larger than the current one. By scale, I mean the coverage of different SSL methods and AE methods, the scale of the dataset, and the architecture. Because what is still unknown to the community is the limit of scaling up in using SSL or AE in domain adaptation/generalization/distribution shifts.
>
>
>
> We believe the experiments in our work are sufficient and thorough. This is echo-ed by reviewer bhk8 and xv8k, who consider our experiments to be comprehensive and extensive. More specifically we covered:
>
> 1) 8 algorithms: including 3 SSL, 4 AE and 1 SL method;
> 2) 3 architectures: CNN, ResNet-50, DenseNet-121;
> 3) Synthetic and realistic distribution shift datasets with varying levels of shift: MNIST-CIFAR, CdSprites, Camelyon17, FMoW.
>
> We are glad that the reviewer agrees that this is an important area of research that needs further investigation, however with the limited amount of computational resources available to us, we would rather investigate a well-defined problem thoroughly, drawing convincing conclusions in a smaller scope, than conducting insufficient experiments on a large scale. We leave the interesting research questions the reviewer raises (larger scale experiments and other modalities beyond images) to future work, and will mention them in our updated manuscript.

---

> > ### Author Response · Authors · 2022-11-08
> > **Thank you for your reviews! (2/2)**
> >
> > > Q4. On the other hand, I think the elephant in the room is still unknown: To what extent can we really rely on the pretrained representations to do the transfer and adaptation? What is the limit?
> >
> > *(Clarification: we would like to repeat our point in Q1 that our work focus more on the distribution shift setting; the transfer learning setting where the model is first pre-trained on large scale data, then adapted to do a different task is not the one under investigation)*
> >
> > The conclusion of our paper is clear:
> > 1. Unsupervised objectives can effectively generalise to unseen distributions, especially when the distribution shift is more extreme;
> > 2. Re-training linear head on OOD data is very important when measuring the generalisation performance of representation learning algorithms.
> >
> > "To what extent can we rely on these conclusions" is a great question to ask, especially for papers that try to understand model's behaviour by performing empirical study. This is the question we asked ourselves many times when designing our experiments, and is exactly why we 1) use multiple algorithms in each style of objectives, 2) study both realistic and synthetic distribution shift settings and 3) propose metrics such as shift sensitivity and linear head bias. In these various experiments, we find the two above conclusions to be true across board. It is difficult to provide a theoretical guarantee that can extend to practical scenarios, however we believe that through our careful experimental design and thorough empirical evaluation, the conclusions above should generalise to image-based distribution shift settings.
> >
> >
> > > Q5. The way we set up the problem may also have some impact on the results. For example, the way we "control" the shift in the images may not be generalizable to real-world shifts in, for example, fairness problems/subpopulation shifts, where we cannot easily create shifts in a meaningful way. So, I also suggest adding discussions on the limitations of the work.
> >
> > We would like to note that our work is the first attempt to create subpopulation shifts from domain generalisation datasets (also see comments by reviewers bhk8, xv8k), and serves as a good starting point to bridge gap between synthetic and real datasets. However we would agree that there are limitations to this method, we will make sure to discuss this in our updated manuscript.
> >
> >
> >
> >
> >
> > > Q6. In table 4, why the SL accuracy is much higher?
> >
> > This is indeed an important question, and is our motivation in studying controllable shift. We discuss this in Section 3.2.1 paragraph 4 and Section 3.2.2 paragraph 1 (after "findings"), which we summarise as follow:
> >
> > The fact that SL is the worst performing model on synthetic datasets (MNIST-CIFAR, CdSprites), but at the same time, one of the best performing models on realistic datasets (FMoW and Camelyon17) highlights the need to studying realistic datasets along with synthetic ones. We hypothesise that this discrepency in performance is because the distribution shift in realistic datasets are more subtle, and that if we increase the intensity of the shift we might observe SL's performance to drop. This hypothesis is validated by experiments in Section 3.2.2 with the controllable shift datasets (FMoW-CS and Camelyon17-CS).
> >
> >
> >
> >
> >
> > **References:**
> >
> > [1] Joshua Robinson, Li Sun, Ke Yu, Kayhan Batmanghelich, Stefanie Jegelka, and Suvrit Sra. Can contrastive learning avoid shortcut solutions? (2021)
> >
> > [2] Sagawa, Shiori, et al. "Extending the wilds benchmark for unsupervised adaptation." (2021)

---

### Official Review · Reviewer_xV8K · 2022-10-27

**Confidence:** 3
**Correctness:** 3
**Technical Novelty And Significance:** 3
**Empirical Novelty And Significance:** 3
**Recommendation:** 8

**Clarity, Quality, Novelty And Reproducibility:**

This study is novel to the best of my knowledge although, given its importance, it is possible that previous has conducted similar experiments. The setting where the classifier is trained on the target domain is close to the feature transfer setting in transfer learning. I think the difference here is the types of datasets that are used and the controllable degree of shift between the source and target domain. If previous work has compared SSL, AE, and SL representation learning in a feature transfer setting and showed that SSL learns more transferrable features then the findings of this paper are less surprising.

**Strength And Weaknesses:**

The consideration of training the classifier head on the target distribution is reasonable and important in my opinion. In domain shift tasks the source domain is generally of a much larger scale than the target task. It is then reasonable to assume that the representation is trained on the source domain and the classifier (which needs less data if it is given a high quality representation) is trained on the smaller target domain dataset.

The set of algorithms and tasks are sufficiently diverse. There is ample evidence for all the claims in the paper. The only exception is the "Findings" paragraph in Section 3.2.2. Figure 3 is supposed to back this claim but the plots show high levels of noise and mixed patterns. For example, in fig 3 (a, top) the error bars mostly overlap and in Fig 3 (b, bottom) SL shows a drop in sensitivity. I suggest adding more runs, more levels of r_id, and clarifying the number of runs and what error bars represent.

I would also like to see a detailed discussion on the augmentation methods used for SSL as this is critical to the performance of the representation learning method. What augmentation methods are used? For each pair of images concatenated together, is the same augmentation method used for both images? Are the two images modified exactly the same way (e.g. if both are rotated, are both images rotated in the same direction and by the same angle)?

Minor comment: The first sentences of the first two bullet points at the end of the introduction are worded too close to each other. I understand that the first point refers to accuracy and the second point refers to shift sensitivity but this is not what those two sentences convey.

**Summary Of The Paper:**

The submission provides an extensive empirical study on supervised (SL), unsupervised (autoencoder, AE), and self-supervised (SSL) representation learning algorithms in a distribution shift scenario. The key contribution is comparing these algorithms when the representation is learned on the source task while the classifier is learned on the target task. The findings are that (1) SL is more sensitive to distribution shift than SSL and AE, and (2) there is a remarkable difference between the performance of classifier heads trained on source and target distributions.

**Summary Of The Review:**

I'm voting for accept as the claims in the paper are clearly stated and well motivated and supported. I elaborated on these points in the Strengths/Weaknesses section and provide a few suggestions.

---

> ### Author Response · Authors · 2022-11-08
> **Thank you for your reviews!**
>
> We thank you for the detailed and thorough review and the very helpful suggestions!
>
> > Q1. The only exception is the "Findings" paragraph in Section 3.2.2. Figure 3 is supposed to back this claim but the plots show high levels of noise and mixed patterns. For example, in fig 3 (a, top) the error bars mostly overlap and in Fig 3 (b, bottom) SL shows a drop in sensitivity. I suggest adding more runs, more levels of r_id, and clarifying the number of runs and what error bars represent.
>
> Figure 3 could indeed need extra clarifications, thank you for your helpful  suggestions! We will make changes to our manuscript, in the meantime we hope the following answers the reviewer's questions:
>
> 1. **Experiment set-up and notation:** Following the guidelines from WILDS benchmark, we perform 10 random seed runs for all Camelyon17 experiment and 3 random seed runs for FMoW. The error bars in all images represent standard deviation.
> 2. **Error bar overlap for Camelyon17 (Fig 3a, top):** The large variance in performance between random trials is a known property of the Camelyon17 dataset [1]. This is why the WILDS benchmark asks for 10 random experiments for this dataset, while only 3 suffices for most others (e.g. FMoW). It is therefore not too surprising that there should be some overlap, however it is still clear that consistent to other experiments, the average accuracy of SL here trends lower than SSL and AE, especially when the distribution shift becomes more extreme.
>
> > Q2. I would also like to see a detailed discussion on the augmentation methods used for SSL as this is critical to the performance of the representation learning method. What augmentation methods are used? For each pair of images concatenated together, is the same augmentation method used for both images? Are the two images modified exactly the same way (e.g. if both are rotated, are both images rotated in the same direction and by the same angle)?
>
> Thank you! Choices of augmentations are indeed crucial for the performance of SSL models, we will be sure to add these details to our paper.
>
> In answer to your question, for all experiments we follow the standard set of augmentations established in [1] and implemented in [2], including random resize crop, random color jittering, random grayscale, random Gaussian blur, random solorisation and random horizontal flip. An exception is the CdSprites experiment where we remove color jittering, as color classification is one of the tasks we are interested in and color jittering would add noise to the labels.
>
> For MNIST-CIFAR, we independently apply random augmentation to MNIST and CIFAR respectively (drawn from the same set of augmentations as detailed above) and then concatenate them to construct training examples. As such the two images are not modified exactly the same way.
>
>
> > Q3. Minor comment: The first sentences of the first two bullet points at the end of the introduction are worded too close to each other.
>
> Thank you for your suggestions, we will reword this in our updated manuscript.
>
>
> **References:**
>
> [1] Xinlei Chen, Haoqi Fan, Ross Girshick, Kaiming He. Improved Baselines with Momentum Contrastive Learning. (2020)
>
> [2] Victor Guilherme Turrisi da Costa, Enrico Fini, Moin Nabi, Nicu Sebe and Elisa Ricci. solo-learn: A Library of Self-supervised Methods for Visual Representation Learning. (2022)

---

> > ### Comment · Reviewer_xV8K · 2022-11-27
> > **Thanks**
> >
> > I went through the other reviews and I'm still voting for accept. While previous findings in feature transfer would imply the conclusions of this paper, this submission serves as a systematic verification of this conclusion. Further, the benchmarks in the submission are clearly different from common feature transfer benchmarks and the submission addresses a misconception about the strength of representation learning methods under domain shift.

---

### Official Review · Reviewer_Bhk8 · 2022-11-03

**Confidence:** 3
**Clarity, Quality, Novelty And Reproducibility:** The paper is clear and the writing is…
**Correctness:** 4
**Technical Novelty And Significance:** 2
**Empirical Novelty And Significance:** 2
**Recommendation:** 6

**Strength And Weaknesses:**

Strength:
- The experiments are very comprehensive and implementation details are provided.
- The construction of the controllable distribution shift dataset is novel.
- As far as I know, this is the first work that explicitly takes retraining the linear head into consideration, and compare its performance side by side with using the ID linear head. This could be an important point that is worth highlighting to the community.

Weakness
- There's limited explanation about why SSL/AE works better than SL on shifted distribution beyond what people already know in the literature.
- To some extent, I feel there's nothing surprising about the findings in this paper. Although it's always nice to have more comprehensive experiments, it's unclear what's the conceptual insight that a reader can gain from this paper.

**Summary Of The Paper:**

This paper provides a comprehensive empirical study of the OOD generalization performance of SSL, AE and SL. In addition to testing standard algorithms on standard distribution shift datasets, the paper proposes a new controllable realistic distribution shift task, and also consider a setting where the linear head is retrained on the OOD data. Results suggest that SSL and AE are more robust to distribution shift than SL.

**Summary Of The Review:**

I think this paper provides a reasonably good empirical comparison between SSL/AE and SL in various settings with different datasets. Although the result doesn't sound surprising, it's good to have this as an addition to the community, hence I would love to recommend for its acceptance.

---

> ### Author Response · Authors · 2022-11-08
> **Thank you for your reviews!**
>
> We thank the reviewer for the insightful comments, and are glad that the reviewer considers our work to be a good addition to the community. Below we try to address some novelty concerns made by the reviewer and welcome any further discussion.
>
>
> > Q1. There's limited explanation about why SSL/AE works better than SL on shifted distribution beyond what people already know in the literature.
>
> In section 2 of our paper we provide the following explanation on why SSL/AE works better than SL on distribution shift:
>
> 1. **Target-driven objective of SL:** when presented with two features that are equally predictive of the target label, SL models have no incentive to learn both as learning only one of them suffices to predict the target label. As highlighted in recent research [1, 2], this property of SL could exacerbate the model's vulnerability to spurious correlations, resulting in poor generalisation performance;
> 2. **Input-driven objective of SSL/AE:** on the other hand, SSL and AE methods are incentivised to learn representations that most accurately represent the input data [3, 4]. When presented with two features equally predictive of the labels, unsupervised learning algorithms encourage the model to go beyond learning what’s enough to predict the label, and instead focus on maximising the mutual information between the representations and the input.
>
>
> We believe that summarising and reasoning using prior work as above suffice in providing theoretical insights to what we observe.
>
>
> > Q2. To some extent, I feel there's nothing surprising about the findings in this paper. Although it's always nice to have more comprehensive experiments, it's unclear what's the conceptual insight that a reader can gain from this paper.
>
>
> The main conceptual insight of our work is that unsupervised learning objectives exhibit stronger generalisation performance, especially under extreme distribution shift. While there are theoretical insights that provide foundations for this finding [1,2,3,4], the benefits of using unsupervised objectives for distribution shift are not widely recognised and utilised. As an example, the popular domain generalisation benchmark DomainBed [5] features 27 SOTA methods that are **all developed for the supervised learning setting**. Motivated by this observation, our work serves to establish the strong generalisation capability of unsupervised objectives through comprehensive, systematic empirical analysis, and we wish to inspire future works on generalisation to shift focus from supervised objectives to unsupervised objectives.
>
> **References:**
>
> [1] Harshay Shah, Kaustav Tamuly, Aditi Raghunathan, Prateek Jain, and Praneeth Netrapalli. The pitfalls of simplicity bias in neural networks. (2020)
>
> [2] Robert Geirhos, Jörn-Henrik Jacobsen, Claudio Michaelis, Richard Zemel, Wieland Brendel, Matthias Bethge, and Felix A Wichmann. Shortcut learning in deep neural networks. (2020)
>
> [3] Ting Chen, Simon Kornblith, Mohammad Norouzi, and Geoffrey E. Hinton. A simple framework for contrastive learning of visual representations. (2020)
>
> [4] Alexander A. Alemi, Ian Fischer, Joshua V. Dillon, and Kevin Murphy. Deep variational information bottleneck. (2017)
>
> [5] Gulrajani, Ishaan, and David Lopez-Paz. "In search of lost domain generalization." (2020)

---

> > ### Comment · Reviewer_Bhk8 · 2022-11-27
> > **Thanks for your response**
> >
> > I agree that these interpretations of ssl vs sl makes sense, but still I’m not convinced that this paper provides enough conceptual novelty. That being said, the empirical results might be a nice contribution to the community. Thus, I’d like to maintain my score and vote for acceptance of the paper.

---

### Author Response · Authors · 2022-11-08
**Summary and General Response**

We thank all the reviewers for their helpful comments, and will adopt suggestions made by xv8k and ekj8 on changes that will improve the clarity and presentation of the paper.

For convenience of further discussion, below we summarise and address the two main concerns raised by reviewers.

1. **On the importance of evaluating unsupervised learning objectives under distribution shift:**
Some reviewers remarked (bhk8, ekj8) that the finding that SSL/AE are better than SL under distribution shift is unsurprising -- while there are theoretical insights that provide foundations for this finding [1,2,3,4], the benefits of using unsupervised objectives for distribution shift task are not widely recognised and utilised. As an example, the popular domain generalisation benchmark DomainBed [5] features 27 SOTA methods that are **all developed for the supervised learning setting**. Motivated by this observation, our work serves to establish the strong generalisation capability of unsupervised objectives through comprehensive, systematic empirical analysis (xv8k, jb4r), and we wish to inspire future works on generalisation to shift focus from supervised objectives to unsupervised objectives.

2. **On the novelty of re-training linear head on OOD data:**
While some reviewers (xv8k, bhk8) list our contribution of re-training linear head on OOD data as one of the main strengths of our paper, reviewer jb4r and ekj8 both raised concerns on the novelty of this set-up. We believe the following two points should help clarify this:

    1) Both reviewers mention that in transfer learning, re-training the linear classifier head on target data is standard practice. However, our work focuses on the distribution shift setting, not transfer learning. This is important because the re-training linear head is **necessary** in transfer learning as the target data typically features a different task than the source data that the model is trained on; in distribution shift however, the source and target data feature the same task -- re-training linear classifier on target domain is therefore **not necessary**, and thus not at all standard practice in distribution shift. However, we show in our experiment that re-training linear classifier significantly improves generalisation performance even on realistic distribution shift tasks, which is a novel finding.
    2) Prior work [6] and [7] also investigates the performance of SSL on distribution shift tasks. However these works did not take the linear head bias into account, and concluded that representations learned from SSL are worse at generalisation. By re-training the linear head, surprisingly, we came to the opposite conclusion to these prior works. This demonstrates the importance of removing the linear head bias when evaluating generalisation performance.

**References:**

[1] Harshay Shah, Kaustav Tamuly, Aditi Raghunathan, Prateek Jain, and Praneeth Netrapalli. The pitfalls of simplicity bias in neural networks. (2020)

[2] Robert Geirhos, Jörn-Henrik Jacobsen, Claudio Michaelis, Richard Zemel, Wieland Brendel, Matthias Bethge, and Felix A Wichmann. Shortcut learning in deep neural networks. (2020)

[3] Ting Chen, Simon Kornblith, Mohammad Norouzi, and Geoffrey E. Hinton. A simple framework for contrastive learning of visual representations. (2020)

[4] Alexander A. Alemi, Ian Fischer, Joshua V. Dillon, and Kevin Murphy. Deep variational information bottleneck. (2017)

[5] Gulrajani, Ishaan, and David Lopez-Paz. "In search of lost domain generalization." (2020)

[6] Joshua Robinson, Li Sun, Ke Yu, Kayhan Batmanghelich, Stefanie Jegelka, and Suvrit Sra. Can contrastive learning avoid shortcut solutions? (2021)

[7] Sagawa, Shiori, et al. "Extending the wilds benchmark for unsupervised adaptation." (2021)

---

### Author Response · Authors · 2022-11-16
**Rebuttal Revision**

We would like to thank the reviewers again for the helpful suggestions to our paper. **Please do let us know if you have any further concerns regarding our work.**

Following reviewers' advice, we have made following changes to the paper to improve its clarity.

- **[bhk8, ekj8]** Highlight the novelty and significance of evaluating unsupervised objectives for distribution shift tasks:
    > Added to bottom of page 3: "Note that though the intuition that unsupervised objectives should be better at distribution shift tasks is well-established in theory (Chen et al. (2020); Alemi et al. (2017)), state-of-the-art methods are predominantly developed under SL. To the best of our knowledge, we are the first to systematically evaluate and compare unsupervised representation learning methods to SL under distribution shift."

- **[bhk8, jb4r, ekj8]** Comparing retraining linear head in distribution shift vs. transfer learning:
    > Added to the end of Problem 3 in Section 2: "Note that although retraining linear classifier head is already standard practice in transfer learning, its application is necessary as the pre-training task and target task are typically different; on the other hand, retraining linear head is neither necessary nor standard practice in distribution shift problems, despite the recognition of linear head bias in recent work (Kang et al., 2020; Menon et al., 2020)."
- **[xv8k]** Add discussions on augmentations used:
    > 1) Section 3, first paragraph: added  "We use the standard SSL augmentations proposed in He et al. (2020); Chen et al. (2020b) for all models to ensure a fair comparison. See appendix B for details."
    > 2) Appendix B, first paragraph: added "For all experiments we follow the standard set of augmentations proposed in He et al. (2020); Chen et al. (2020b), including random resize crop, random color jittering, random grayscale, random Gaussian blur, random solorisation and random horizontal flip. An exception is the CdSprites experiment where we remove color jittering, as color classification is one of the tasks we are interested in and color jittering would add noise to the labels. For MNIST-CIFAR, we independently apply random augmentation to MNIST and CIFAR respectively (drawn from the same set of augmentations as detailed above) and then concatenate them to construct training examples."
- **[xv8k]** Reword the first two contribution bullet points to highlight their difference:
    > Reworded second bullet point, the two bullet points are now:
    > - SSL and AE are more robust than SL to extreme distribution shift;
    >- Compared to SL, SSL and AE’s performance drop less under distribution shift.
- **[xv8k]** Clarifications on Figure 3:
    > 1) Added to Section 3.2, first paragraph: "Following the guidelines from WILDS benchmark, we perform 10 random seed runs for all Camelyon17 experiment and 3 random seed runs for FMoW. The error bars in Figure 3 represent standard deviation. "
    > 2) Added to Section 3.2, fourth paragraph: "Note that the standard deviation for all three methods are quite high for Camelyon17: this is a known property of the dataset and similar pattern is observed across most methods on WILDS benchmark (Koh et al., 2021)."
- **[ekj8]** Reorder Figure 1 and Figure 2 to avoid forward referencing;
- **[ekj8]** Typo fixes:
    > 1) “This is becaues, …” -> “This is because,…”
    > 2) “under distribution shif, …” -> “under distribution shift, …”

---

### Decision · Program_Chairs · 2023-01-20

**Decision:**

Accept: poster

**Justification For Why Not Higher Score:**

The overall novelty of the paper is not very high, but it does contain observations and evaluations that would be of interest to the community. Presentation at poster level seems adequate.

**Justification For Why Not Lower Score:**

There was initially one low score, which was increased, and upon reading the current reviews, where one of the 5 weakly supports accepting, I think it is ok to accept the paper.

**Metareview: Summary, Strengths And Weaknesses:**

The paper presents a systematic study of learning under distribution shift. The reviewers appreciated the empirical setup where distribution shift is generated in a controlled manner. They also thought the empirical conclusions (of retraining the linear head on OOD data). While there is no dramatically new contribution here, the overall view of the reviewers is that this would be useful for the community to know.

**Note From Pc:**

if the above contains the word "oral" or "spotlight" please see: "oral" presentation means -> notable-top-5% and "spotlight" means -> notable-top-25%. As stated in our emails, we are disassociating presentation type from AC recommendations